# Nutritional compensation of the circadian clock is a conserved process influenced by gene expression regulation and mRNA stability

Christina M. Kelliher [1,2]*, Elizabeth-Lauren Stevenson[1], Jennifer J. Loros[3], Jay C. Dunlap[1]*

1 Department of Molecular & Systems Biology, Geisel School of Medicine at Dartmouth, Hanover, New Hampshire, United States of America, 2 Department of Biology, University of Massachusetts Boston, Boston, Massachusetts, United States of America, 3 Department of Biochemistry & Cell Biology, Geisel School of Medicine at Dartmouth, Hanover, New Hampshire, United States of America

* christina.kelliher@umb.edu (CMK); jay.c.dunlap@dartmouth.edu (JCD)

**Data Availability Statement:** RNA-Sequencing gene expression data from this manuscript have been submitted to the NCBI Gene Expression Omnibus (GEO; http://www.ncbi.nlm.nih.gov/geo/)

## Abstract

Compensation is a defining principle of a true circadian clock, where its approximately 24-hour period length is relatively unchanged across environmental conditions. Known compensation effectors directly regulate core clock factors to buffer the oscillator's period length from variables in the environment. Temperature Compensation mechanisms have been experimentally addressed across circadian model systems, but much less is known about the related process of Nutritional Compensation, where circadian period length is maintained across physiologically relevant nutrient levels. Using the filamentous fungus *Neurospora crassa*, we performed a genetic screen under glucose and amino acid starvation conditions to identify new regulators of Nutritional Compensation. Our screen uncovered 16 novel mutants, and together with 4 mutants characterized in prior work, a model emerges where Nutritional Compensation of the fungal clock is achieved at the levels of transcription, chromatin regulation, and mRNA stability. However, eukaryotic circadian Nutritional Compensation is completely unstudied outside of *Neurospora*. To test for conservation in cultured human cells, we selected top hits from our fungal genetic screen, performed siRNA knockdown experiments of the mammalian orthologs, and characterized the cell lines with respect to compensation. We find that the wild-type mammalian clock is also compensated across a large range of external glucose concentrations, as observed in *Neurospora*, and that knocking down the mammalian orthologs of the *Neurospora* compensation-associated genes *CPSF6* or *SETD2* in human cells also results in nutrient-dependent period length changes. We conclude that, like Temperature Compensation, Nutritional Compensation is a conserved circadian process in fungal and mammalian clocks and that it may share common molecular determinants.

under accession number GSE201901. All other relevant data are within the paper and its Supporting information files.

**Funding:** This work is supported by the National Institutes of Health (NIH) grants # F32 GM128252 to CMK (https://reporter.nih.gov/project-details/10513105), R35 GM118021 to JCD (https://reporter.nih.gov/project-details/10330086), and R35 GM118022 to JJL (https://reporter.nih.gov/project-details/9902458). The funders had no role in study design, data collection and analysis, decision to publish, or preparation of the manuscript.

**Competing interests:** The authors have declared that no competing interests exist.

**Abbreviations:** AMP, adenosine monophosphate; APA, Alternative Polyadenylation; CKI, Casein Kinase I; CPSF6, Cleavage and Polyadenylation Specificity Factor subunit 6; FGSC, Fungal Genetics Stock Center; GEO, Gene Expression Omnibus; GO, Gene Ontology; GSR, Genomics Shared Resource; MEF, mouse embryonic fibroblast; NAD, nicotinamide adenine dinucleotide; NMD, nonsense-mediated decay; RT-qPCR, quantitative reverse transcription PCR; SE, single-end; TTFL, transcription-translation feedback loop; WCC, White Collar Complex.

## Introduction

Circadian clocks exist at the cellular level to allow cell types, tissues, and organisms to properly align physiology with time of day. True circadian clocks are sensitive to the external environment in two distinct ways. Discrete pulses of bright light, temperature, nutrients, hormones, or other chemicals reset circadian oscillators and reorient the clock's phase to the new environment (reviewed in [1]). This resetting feature of circadian clocks is most commonly experienced when jetlagged humans travel across multiple time zones and entrain to the destination's light/dark cycles. On the other hand, circadian clocks are also shielded or buffered from changes in the ambient environment within the physiological range of an organism [2,3]. In the filamentous fungus *Neurospora crassa*, circadian period length is maintained at approximately 21.5 hours when grown at constant temperatures ranging from 16°C to 32°C [4] or under different nutrient conditions [5]. This circadian property is known as compensation, and the molecular mechanisms underlying period length compensation remain elusive.

At the molecular level, the circadian oscillator is a transcription-translation feedback loop (TTFL) that is functionally conserved from fungi to animals. The positive arm of the clock is composed of transcription factor activators (WC-1/WC-2 in fungi; BMAL1/CLOCK in mammals), which form a heterodimeric complex, drive transcription of direct target genes, and recruit chromatin modifiers [6–8]. The positive arm directly regulates the negative arm of the clock (FRQ in fungi; PERs/CRYs in mammals) [9–11]. Negative arm clock components form a stable complex with Casein Kinase I (CKI) and other factors, leading to feedback via posttranslational inhibition of the positive arm to close the circadian feedback loop [12,13]. The positive and negative arms are sufficient for rhythmicity, although accessory feedback loops confer additional clock robustness (reviewed in [14]). Such individual cellular clocks are coordinated in a coupled network to align organismal physiology in mammals (reviewed in [15]).

Temperature Compensation was first proposed to be a circadian property in the dinoflagellate *Gonyaulax polyedra* when increasing temperatures led first to period lengthening (so-called "over-compensation") and then to period shortening at even higher temperatures, a result that plainly conflicted with models based on biochemical reaction rates strictly increasing with temperature [2]. In genetic model systems like *Neurospora* and *Drosophila* where organisms operate at ambient temperatures, and even in homeothermic animals, cellular circadian clocks are temperature compensated [4,16–20]. The forward and reverse genetics that have driven current models for Temperature Compensation have led to casein kinases as central regulators across multiple circadian model systems. In *Neurospora* and in plants, Casein Kinase II (CKII) is required for Temperature Compensation [21,22]. *Neurospora* CKII activity increases linearly with temperature and directly phosphorylates the negative arm of the clock [21]. The *tau* mutant hamster (CKIe$^{R178C}$) was the first characterized mammalian Temperature Compensation defect [23–25]. Recent structural work has demonstrated that CKI *tau* alters both priming and progressive phosphorylation events on the negative arm of the clock [26], and a *Neurospora tau*-like allele CKI$^{R181Q}$ reduces in vitro kinase activity on phospho-primed substrates [27]. Taken together, the *tau* mutant phenotype of period shortening with temperature (so-called "under-compensation") is likely due to altered progressive phosphorylation, and perhaps to altered protein stability, of negative arm components. Indeed, temperature-sensitive target binding has been shown for mammalian CKI [28], and the interaction strength between CKI and FRQ in *Neurospora* has been implicated in Temperature Compensation as well [29]. Taken together, CKI and CKII each have temperature-sensitive aspects of enzyme activity, both can directly phosphorylate core clock components, and casein kinases appear to play a conserved role in Temperature Compensation of fungal, plant, and animal clocks.

In contrast to Temperature Compensation, mechanisms underlying Nutritional Compensation (also known as Metabolic or Glucose Compensation) are poorly understood in *Neurospora* and completely unstudied in other eukaryotic circadian systems. Period compensation to a variety of ATP:ADP ratios has been well described in the prokaryotic cyanobacteria *Synechococcus elongatus* (reviewed in [30]). A handful of Nutritional Compensation defects have arisen sporadically in *Neurospora*, the most developed involving the transcription factor repressor CSP-1 [31]. CSP-1 directly regulates and is regulated by the clock's positive arm White Collar Complex (WCC), forming an accessory negative feedback loop. In a Δ*csp-1* mutant, period significantly shortens as a function of glucose concentration [32]. In fact, direct overexpression of *wc-1* also causes nutritional under-compensation [33]. Nutritional Compensation is defective in the absence of the general transcription repressor RCO-1 [34], likely due its normal role in preventing WCC-independent *frq* transcription [35] and/or its dual repressor function in complex with CSP-1 [36]. Over-compensation was also found in loss-of-function mutants of an RNA helicase, PRD-1, which normally localizes to the nucleus only under high-glucose conditions [37]. Nutrient sensing and signaling pathways should presumably also play a role in Nutritional Compensation of the clock, and RAS2 and cAMP signaling have been implicated [38]. Taken together, the current incomplete model for Nutritional Compensation in *Neurospora* assembled from random hits implicates transcriptional and posttranscriptional regulation of core clock factors by transcription factors, an RNA helicase, and cAMP signaling.

Since the realization of circadian compensation, the field has speculated that Temperature and Nutritional Compensation pathways may share common regulators [30,39,40]. We directly test this model and find in *Neurospora* that previously reported compensation mutants are specific to either Temperature or Nutritional Compensation. Given this separation of function and relatively little mechanistic knowledge about Nutritional Compensation, we designed a genetic screen to identify new compensation mutants in *Neurospora crassa*. We identify 16 gene knockouts with Nutritional Compensation phenotypes, greatly expanding the list of 4 previously characterized mutants. We also provide the first evidence that, like Temperature Compensation, Nutritional Compensation is relevant in the mammalian circadian system, and confirm that knockdowns of the human genes homologous to two hits from our genetic screen, *CPSF6* and *SETD2*, impair the clock's ability to maintain its period length at different glucose levels. These data establish a potentially conserved genetic basis for the phenomenon of circadian Nutritional Compensation and anchor the phenomenon for further genetic and molecular dissection.

## Results

### Nutritional compensation is distinct from temperature compensation in *Neurospora*

We first set out to characterize the properties of Nutritional Compensation in wild-type *Neurospora*. Traditional circadian experiments have utilized the *ras-1*[bd] mutant, which promotes the formation of circadianly regulated distinct bands of conidial spores in a race tube assay [41,42]. However, the *ras-1* gene (NCU08823) is implicated in growth and regulation of reactive oxygen species. Thus, to accurately profile normal Nutritional Compensation, a wild-type *ras-1*[+] strain containing a *frq* clock box transcriptional reporter was used to measure period length across glucose concentrations ranging from 0 to 111 mM (0 to 2% w/v). The *Neurospora* circadian period is slightly under-compensated to nutrients (Fig 1A); under-compensation has also been observed in wild-type *Neurospora* with respect to temperature [21,29]. Fungal biomass increases by orders of magnitude when grown in the range of 0% to 0.75% w/v glucose

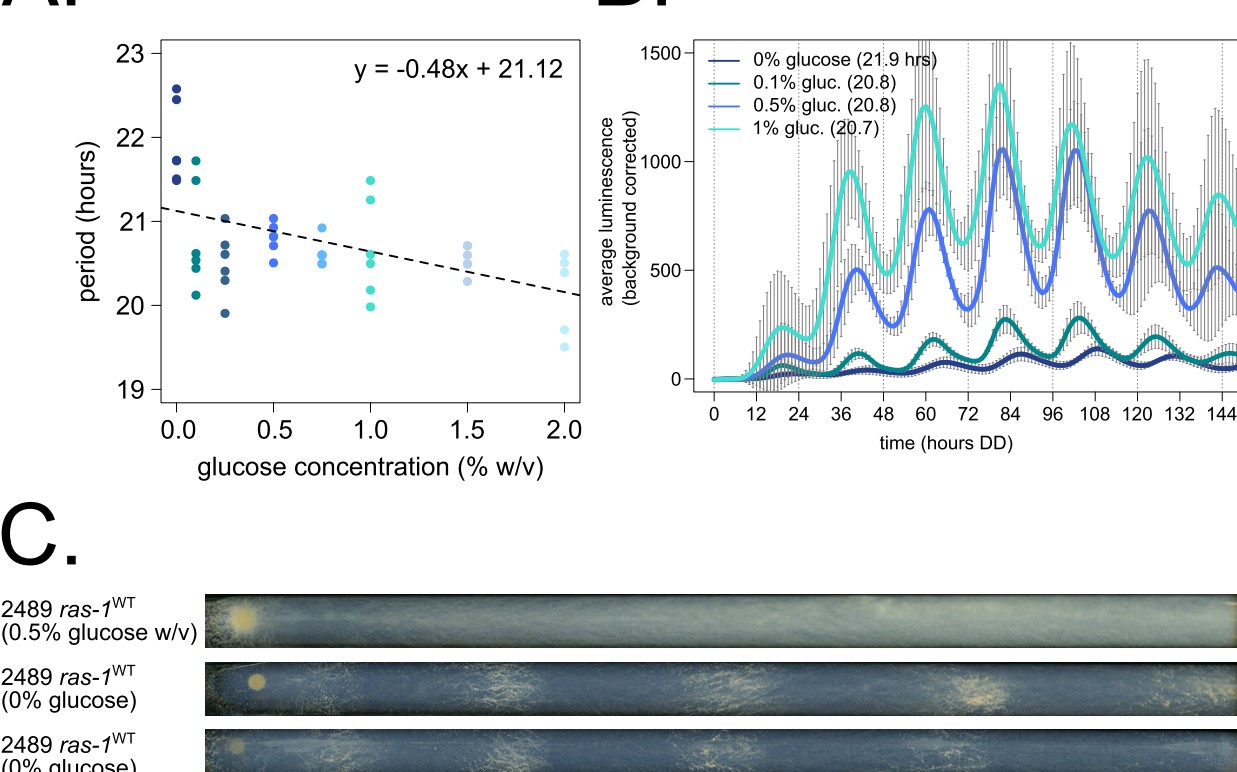

**Fig 1. Nutritional Compensation properties of *Neurospora crassa*.** Circadian period length was determined by bioluminescent recordings of a *ras-1*^WT strain cultured on race tubes, where the fungal growth front encounters constant glucose concentrations ($N = 6$ race tubes per concentration). No arginine was added to the medium. Period shortens slightly as a function of glucose levels, which is indicated by the negative slope of the linear fit (glm in R, Gaussian family defaults, slope = $-0.48 \pm 0.12$) (**A**). Averaged biological and technical replicates are shown for 0 mM (0% w/v), 5.6 mM (0.1% w/v), 27.8 mM (0.5% w/v), and 55.5 mM (1% w/v) glucose levels (standard deviation error bars). Period lengths are $21.9 \pm 0.5$, $20.8 \pm 0.6$, $20.8 \pm 0.2$, and $20.7 \pm 0.6$ hours, respectively (average ± SD) (**B**). Surprisingly, the *ras-1*^WT strain can form distinct conidial bands when grown on 0% glucose starvation medium, contrasted with constitutive conidiation seen at high glucose levels (**C**). The raw data underlying parts (**A**) and (**B**) can be found in S1 Data.

(S1 Fig), and this increased biomass accounts for the increased magnitude of luciferase rhythms observed at higher glucose levels (Fig 1B) (S1 Movie).

Having established that the *Neurospora* clock displays compensation for period length across glucose concentrations, we asked whether Temperature Compensation mutants also have Nutritional Compensation phenotypes, and vice versa, together in the same assay. CKII is required for normal Temperature Compensation, and its catalytic subunit reduced function mutant *cka*^prd-3 (Y43H) has an over-compensation phenotype by race tube assay [21]. The *frq*^7 (G459D) point mutant is temperature under-compensated [4]. The *frq* clock box transcriptional reporter was integrated into the *cka*^prd-3 and *frq*^7 mutant backgrounds, and temperature over- ($Q_{10} = 0.88$) and under-compensation ($Q_{10} = 1.11$) phenotypes were confirmed by luciferase assay (Fig 2A). Known Nutritional Compensation mutants *prd-1* and $\Delta$*csp-1* were also tested, and each had normal Temperature Compensation profiles ($Q_{10} = 1.04$, 1.06). The same 4 compensation mutant reporter strains were then grown on race tubes containing zero or high glucose medium, and bioluminescence was recorded to track period length and Nutritional Compensation phenotypes. Controls were slightly under-compensated

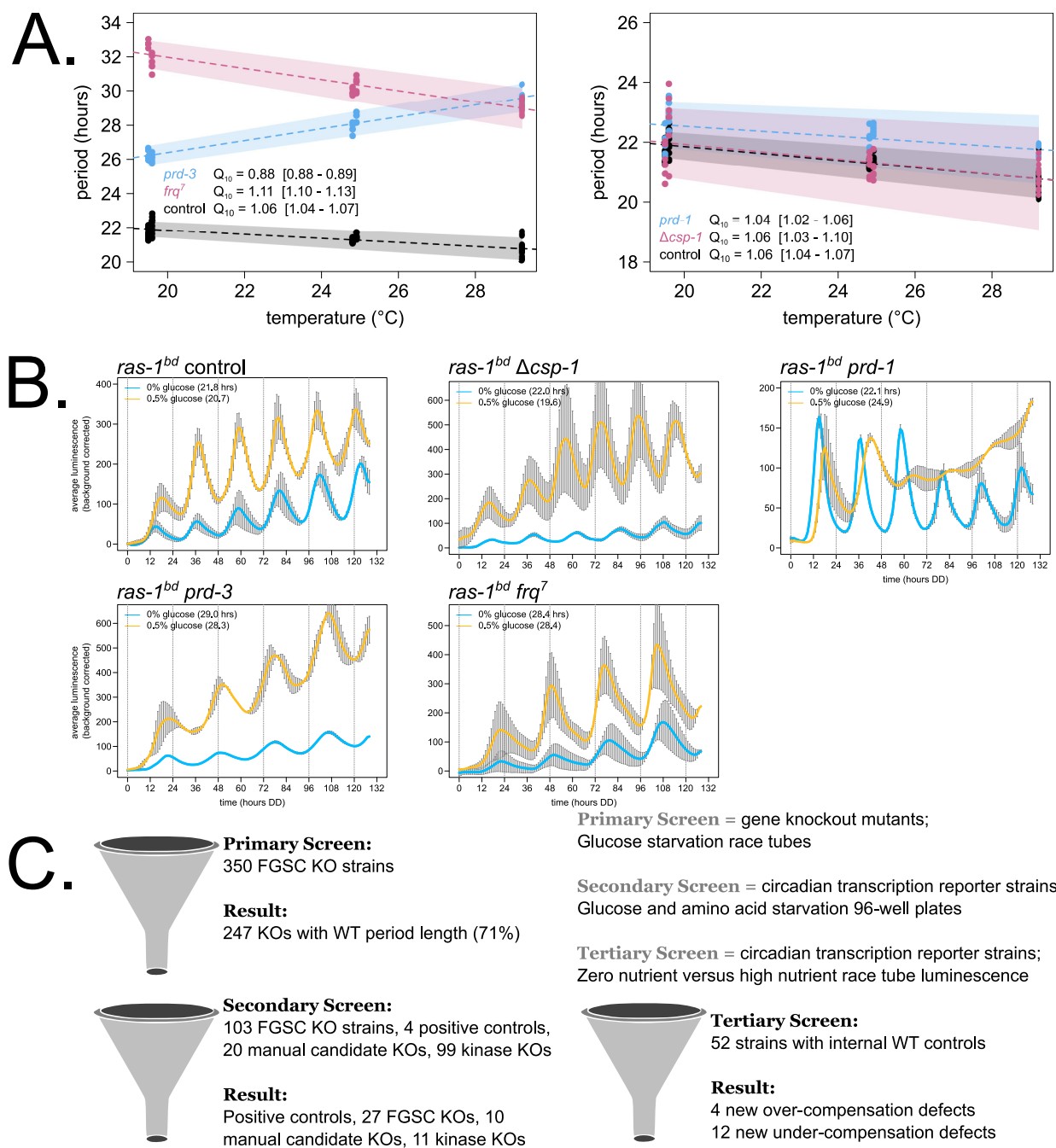

**Fig 2. Nutritional Compensation mutants have normal Temperature Compensation in *Neurospora*, catalyzing a genetic screen for new mutants.** To measure the circadian period length across temperatures in constant darkness, 96-well plate luciferase assays were used ($N = 12$ replicates per strain, per temperature). A linear model was fit to period length data from each strain (glm in R), and a $Q_{10}$ temperature coefficient was calculated using the model-fitted period lengths at 20°C and 30°C. Shaded areas around the linear fit represent the 95% confidence intervals on the slope. Error ranges on $Q_{10}$ values were computed from the 95% CIs (**A**). Circadian bioluminescence was recorded from race tube cultures of the indicated genotypes, as previously described [51] (S1 Movie). High nutrient medium (yellow lines) contained 0.5% w/v glucose 0.17% w/v arginine, and zero nutrient medium (blue lines) contained 0% glucose 0% arginine (standard deviation error bars). Period lengths, indicated as insets, were computed from $N = 2$–5 biological replicates per nutrient concentration (**B**). Cartoon depiction of the 3-phase genetic screen for Nutritional Compensation defects among kinase, RNA regulatory proteins, and manual candidate knockout strains (**C**). FGSC = Fungal Genetics Stock Center, curator of the *Neurospora* whole genome knockout collection. The raw data underlying parts (**A**) and (**B**) can be found in S2 Data.

(Figs 1A and 2B). Nutritional Compensation mutants *prd-1* and *Δcsp-1* showed the over- and under-compensation phenotypes reported by previous studies [32,37]. Temperature Compensation mutants *cka^{prd-3}* and *frq^7* have normal Nutritional Compensation (Fig 2B). We conclude that Temperature and Nutritional Compensation are controlled by distinct pathways in *Neurospora*. These data suggest that further examination of available mutants defective in Temperature Compensation will not inform our understanding of Nutritional Compensation and that a separate genetic screen is warranted.

To achieve this, we leveraged the whole genome knockout collection in *Neurospora crassa* [43] to initiate a screen for new Nutritional Compensation regulators. Two major classes of gene knockouts were selected to search for compensation phenotypes. Kinases are central to many aspects of cellular processes and regulation including responses to the environment (pheromones, osmotic conditions, carbon/nitrogen regulation, etc.), and a collection of approximately 100 different kinase knockout circadian reporter strains was available from previous work [44]. In addition to posttranslational modifications by kinases, posttranscriptional regulation is emerging as critically important for circadian output [45]. Together with the dramatic nutritional over-compensation defect seen in *prd-1* RNA helicase mutants [37] (Fig 2B), we selected 350 putative RNA regulatory protein knockouts to screen for compensation regulators. Our list of putative RNA-binding and RNA regulatory proteins was derived from bioinformatic databases and from the literature [46–49] to include proteins with nucleotide-binding functional annotation but exclude known transcriptional regulators. Multiple circadian period alterations were identified in a recent characterization of transcription factor knockouts [50], and Nutritional Compensation defects among transcription factors, in addition to CSP-1 and RCO-1, will be the subject of future study. Finally, a handful of manually selected candidate genes were included in the compensation screen. Eight classical alleles with long or short circadian period (the series of *prd* mutants and *frq* point mutants) were included, and 12 knockouts were manually selected based on reported roles in nutrient sensing and signaling.

The genetic screen for Nutritional Compensation defects was divided into three phases (Fig 2C). The 350 previously uncharacterized RNA regulatory protein knockout strains were grown on glucose starvation race tubes (Fig 1C) to identify those with period length differences greater than 1 hour as compared to internal wild-type controls or those "output" mutants clearly lacking a circadian banding pattern (*N* = 2 to 8 replicate race tubes per KO strain). Approximately 71% of the putative RNA regulatory protein knockout strains showed normal period length on starvation medium and were eliminated in the primary screen (Fig 2C and S1 Table). During this phase of the screen, 7 knockout strains were found with enhanced conidial banding patterns (and variable growth rates) relative to wild-type controls, reminiscent of the *ras-1^{bd}* mutant phenotype (S2 Fig). In the secondary screen, the *frq* clock box transcriptional reporter was integrated into all knockouts of interest, along with the existing kinase deletion collection [44] and manually selected candidate strains. This collection of more than 200 strains was screened in a high-throughput 96-well plate format using glucose and amino acid starvation medium to identify period length differences greater than 1 hour from internal wild-type controls (*N* = 6 to 12 replicate wells per KO strain). About 77% of the candidate strains were eliminated during the secondary screen due to normal period length on starvation medium (Fig 2C and S1 Table). The approximately 50 remaining knockout strains and wild-type controls were then advanced to the lowest throughput tertiary screen: bioluminescence race tube assays directly comparing zero versus high glucose and amino acid medium (0.5% w/v glucose, 0.17% w/v arginine). It should be noted here that the race tube assay is particularly well suited for a Nutritional Compensation mutant screen because the growth front, which produces most of the bioluminescence, always encounters fresh medium; thus, there is no

complicating effect of nutrient depletion over the course of a six-day assay (S1 Movie). Circadian period lengths were quantified to assay Nutritional Compensation phenotypes. Sixteen new mutants emerged from the genetic screen showing large period changes between zero and high nutrient conditions (Fig 2C and S1 Table).

## mRNA stability and polyadenylation factors emerge as key regulators of circadian period length across nutrients

We first examined the group of 12 new under-compensation mutants, where period length shortens as a function of nutrient levels (Fig 3A and S3 Fig and S1 Table). To prioritize screen hits for further analysis, a Gene Ontology (GO) Enrichment of the 12 hits was performed using FungiDB [47]. Unsurprisingly, the top-scoring GO Term was RNA binding (GO:0003723, BH-adjusted $p = 2.18 \times 10^{-7}$). poly(A) binding was also a top GO Term (GO:0008143, BH-adjusted $p = 1.28 \times 10^{-4}$), and upon further manual inspection of the new under-compensated hits, 4 of the 12 mutants share a common function in polyadenylation of nascent RNAs. NCU02736 (FGSC12857) is designated as an uncharacterized gene in *Neurospora* but is broadly conserved in fungi. Its *Saccharomyces cerevisiae* homolog is a component of the mRNA cleavage and polyadenylation factor I complex (YGL044C, *RNA15*). PABP-2 (NCU03946, FGSC19900) binds in poly(A) tail regions and can broadly regulate mRNA stability. NAB2 (NCU16397, FGSC22799^Het) shares a common domain with the yeast gene *NAB2* (YGL122C), which plays a role in mRNA export and stability. NCU09014 (FGSC12033) is a putative component of a *Neurospora* cleavage and polyadenylation complex and is described in further detail below. The two knockout mutants with the most significant period change between high and zero nutrients are both subunits of the nonsense-mediated decay (NMD) machinery in *Neurospora* (Δupf1^prd-6 and Δupf2) and showed more dramatic nutrient-responsive period shortening than Δcsp-1 [32]. Taken together, regulation of polyadenylation and/or mRNA stability is a common axis of period maintenance with increasing nutrients. The other 6 new under-compensation mutants did not share an obvious functional pathway by GO Term analysis, although some of these genes do show circadian rhythms at the mRNA or protein level and/or are regulated in response to light (S2 Table). However, given the finding that wild-type *Neurospora* is slightly under-compensated (Figs 1A, 2B and 3), the remaining under-compensation mutants (S3 Fig) were not pursued further in favor of functionally related under-compensation mutants and the group of new over-compensation mutants.

The NMD pathway is required for a normal circadian period length due to its regulation of *casein kinase I* mRNA levels [52]. We hypothesized that NMD regulation of *ck-1a* also underlies its nutritional under-compensation defect. NMD in *Neurospora* can be triggered by long 3′ UTR sequences, and the *ck-1a* 3′ UTR is among the longest 1% in *Neurospora* [52]. We generated a *ck-1a* mutant strain lacking its entire 3′ UTR, hypothesizing that this would decouple *ck-1a* transcripts from NMD targeting (Fig 3B). The *ck-1a*Δ3′UTR mutant does indeed have a short period length and shows nutritional under-compensation, phenocopying NMD mutants (Fig 3C). Further, we confirmed that CKIa protein levels are highly up-regulated in both Δupf1^prd-6 and *ck-1a*Δ3′UTR mutant backgrounds (Fig 3D), consistent with our previous work [52]. However, the *ck-1a*Δ3′UTR under-compensation phenotype may not be as severe as observed in NMD mutants, as it displays average period shortening of only 3.5 hours (versus 4.5 hours) in the high nutrient condition (Fig 3C, ANOVA Tukey HSD *p*-values 4 to 5× lower for Δupf1^prd-6 versus *ck-1a*Δ3′UTR). Although the difference between *ck-1a*Δ3′UTR and Δupf1^prd-6 is not statistically significant (ANOVA Tukey HSD *p*-value = 0.057), the additional 1-hour period shortening in Δupf1^prd-6 suggested that multiple NMD targets may be required for normal Nutritional Compensation. Interestingly, NMD mutants also show an

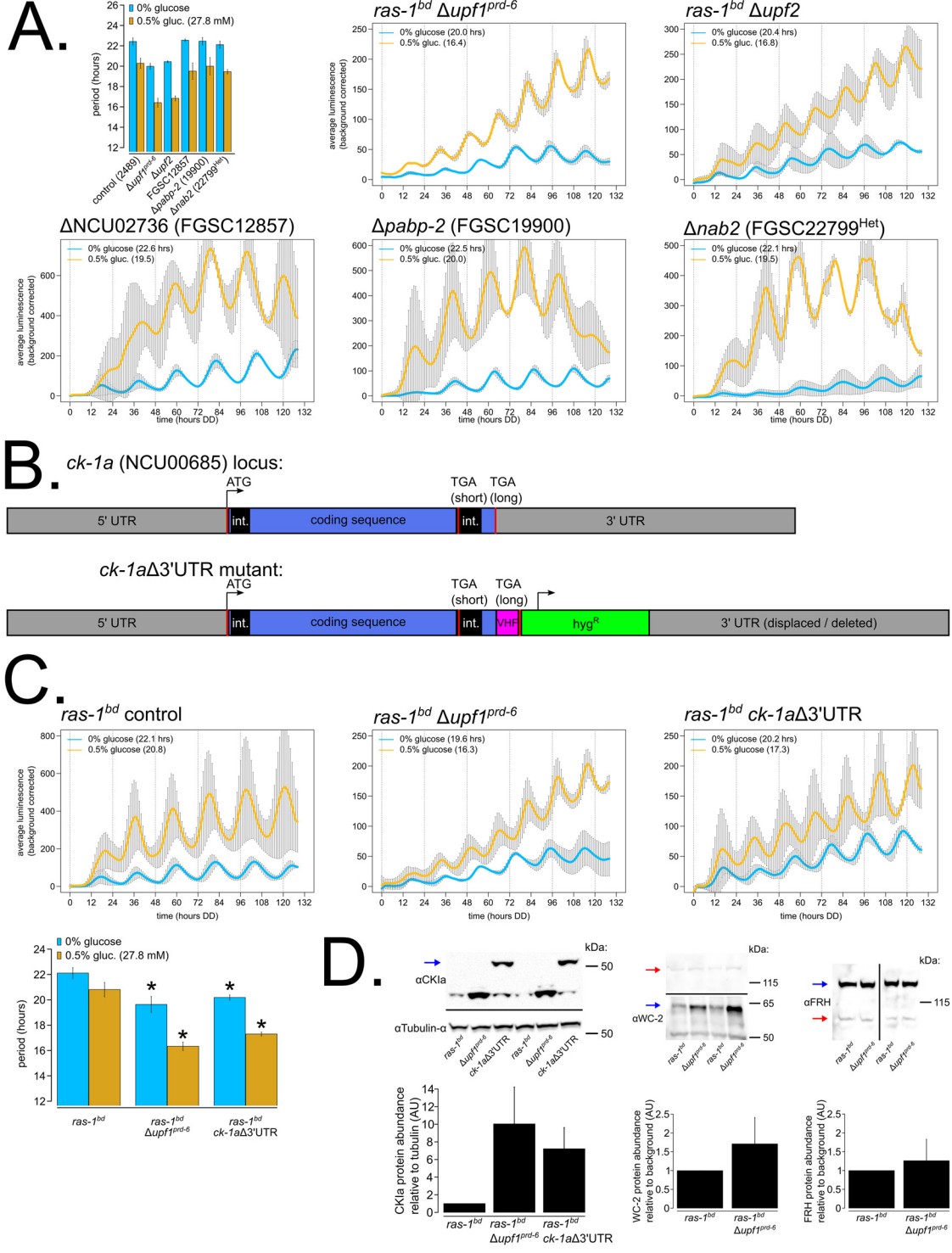

**Fig 3. Nutritional under-compensation mutants are enriched for regulators of mRNA stability, and nonsense-mediated decay (NMD) regulation of the *casein kinase I* 3′ UTR is required for functional Nutritional Compensation and normal period length.** Circadian bioluminescence was recorded from race tube cultures of the indicated deletion mutants. High nutrient medium (yellow lines) contained 0.5% w/v glucose 0.17% w/v arginine, and zero nutrient medium (blue lines) contained 0% glucose 0% arginine (standard deviation error bars). Period lengths were computed ($N \geq 2$ biological replicate period estimates per nutrient concentration) and summarized in a bar graph compared to controls. "Het" indicates heterokaryon strains derived from the *Neurospora* whole genome deletion collection, which were maintained on hygromycin selection medium prior to the bioluminescence race tube assays (**A**). The CKI 3′ UTR mutant was constructed by C-terminal epitope-tagging (V5-10xHis-

3xFLAG) the LONG isoform of NCU00685 at the endogenous locus and displacing 1,543 bps of the annotated 3′ UTR region. Mutant strain construction is shown with a cartoon diagram to scale (**B**). Circadian bioluminescence was recorded from race tube cultures of the indicated genotypes. High nutrient medium (yellow lines) contained 0.5% w/v glucose 0.17% w/v arginine, and zero nutrient medium (blue lines) contained 0% glucose 0% arginine (standard deviation error bars). Period lengths were computed ($N$ = 3–4 biological replicates per nutrient concentration) and summarized in a bar graph. Statistical significance was assessed using ANOVA and Tukey tests (aov & TukeyHSD in R). For zero nutrients (blue bars), the $\Delta upf1^{prd-6}$ (19.6 hours, $p$ = 0.0005, $*$) and $ck$-$1a\Delta3'$UTR (20.2 hours, $p$ = 0.002, $*$) period lengths are significantly shorter than the control (22.1 hours) but no different from each other ($p$ = 0.35). For high nutrients (yellow bars), the $\Delta upf1^{prd-6}$ (16.3 hours, $p$ = $6 \times 10^{-6}$, $*$) and $ck$-$1a\Delta3'$UTR (17.3 hours, $p$ = $3 \times 10^{-5}$, $*$) period lengths are significantly shorter than the control (20.8 hours) but no different from each other ($p$ = 0.06) (**C**). CKIa, WC-2, and FRH protein levels were measured from the indicated genotypes grown on 0.25% w/v glucose + 0.17% w/v arginine high nutrient solid medium for 48 hours in constant light at 25˚C. A representative immunoblot of 2 replicates is shown (out of 4 total biological replicates), and all replicates are quantified in a bar graph relative to $ras$-$1^{bd}$ control protein levels (standard deviation error bars). CKIa expression was normalized to alpha tubulin (NCU09132 and NCU09468) as a loading control, and the blue arrow indicates the higher molecular weight epitope-tagged isoform CKIa$^{LONG}$-VHF (approximately 7 kDa size increase). WC-2 expression (blue arrow) was normalized to a high molecular weight background band (red arrow) from the same blot. FRH expression (blue arrow) was normalized to a lower molecular weight background band (red arrow) from the same blot (**D**). FGSC = Fungal Genetics Stock Center, curator of the *Neurospora* whole genome knockout collection. The raw data underlying parts (**A**), (**C**), and (**D**) can be found in S3 Data.

approximately 1.4-fold increase in *wc-2* and an approximately 1.4-fold decrease in *frh* mRNA expression levels (Figure 5B from [52,53]). *wc-2* is up-regulated in NMD mutants, which could easily explain the under-compensation phenotype because *wc-1* overexpression alone is sufficient to drive nutritional under-compensation [33]. Curiously, we find that *frh*, the obligate binding partner of the disordered protein FRQ [54], is 18-fold down-regulated in response to carbon starvation in wild-type *Neurospora* [55] (S4 Fig). In fact, both *prd-1* and *frh* are among the top 220 genes in the entire *Neurospora* transcriptome that decrease in expression after glucose starvation. WC-2 levels are on average approximately 1.7-fold up-regulated in the $\Delta upf1^{prd-6}$ mutant background, while FRH levels are unchanged (Fig 3D). Thus, period shortening with increasing nutrient levels in NMD mutants (Fig 3A) can be explained by an increase in the mRNA and protein expression of CKIa (Fig 3B–3D) [52] and by higher levels of WC-2 protein (Fig 3D).

Mutations in two genes, NCU02152 (FGSC16956) and NCU09014 (FGSC12033), revealed unique nutritional phenotypes compared to the remaining set of 12 new under-compensation mutants (Fig 3A and S3 Fig). ΔNCU02152 (FGSC16956) had the shortest period phenotype observed in the entire primary and secondary screens (S1 Table). NCU02152 was undescribed in the *Neurospora* literature but contains protein domain homology to the mammalian Cleavage and Polyadenylation Specificity Factor subunit 6 (*CPSF6*). CPSF6 is a member of the CFIm complex, which binds in the 3′ end of nascent mRNAs and facilitates cleavage and poly (A) tail placement. The CFIm complex also contains a second essential component CPSF5. Using the human CPSF5 protein sequence, NCU09014 was confidently identified as its *Neurospora* homolog (reciprocal BLAST e-values = $1e^{-72}/3e^{-74}$). The single mutant ΔNCU09014/Δ*cpsf5* (FGSC12033) had the same short period length as Δ*cpsf6* (S1 Table), which indicates an obligate multimeric complex as with the mammalian CFIm complex [56]. In the tertiary screen, Δ*cpsf5* and Δ*cpsf6* mutants showed progressive period shortening after approximately 60 hours into the circadian free run, specifically when grown on high nutrient medium. Recalling the *prd-1* region-specific Nutritional Compensation phenotype where period defects were seen at the growth front encountering fresh medium, but not at the point of inoculation where nutrients were depleted [37], we checked period length from old/aging tissue surrounding the point of fungal inoculation in Δ*cpsf5* and Δ*cpsf6* mutants. An approximately 18-hour short period length was observed in the old tissue region of Δ*cpsf5* and Δ*cpsf6* mutants grown on high arginine (Fig 4A), compared to an approximately 20-hour period length on zero nutrient medium (S1 Table). This result indicates that Δ*cpsf5* and Δ*cpsf6* mutants must undergo a

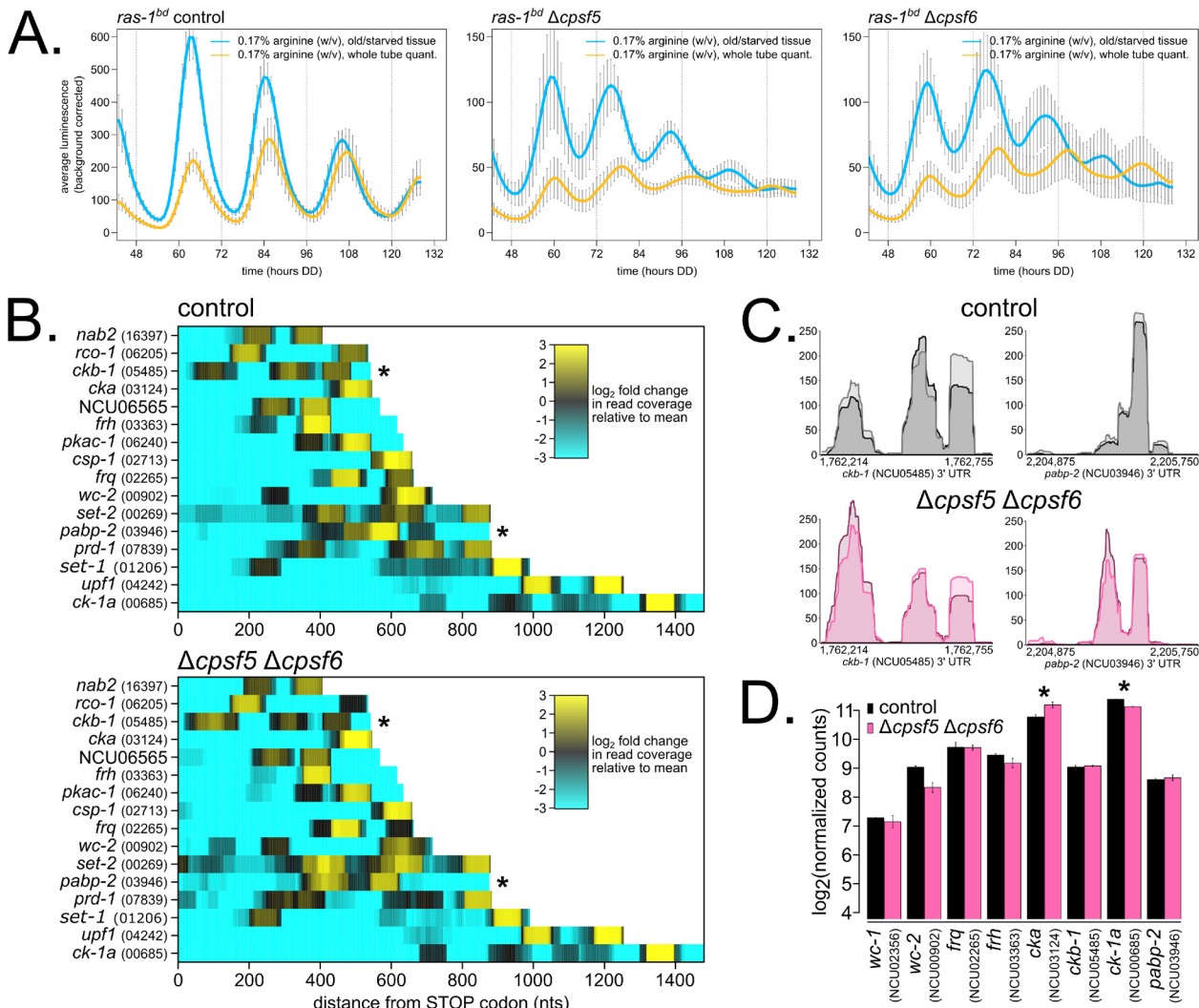

**Fig 4. The *Neurospora* CFIm complex is involved in Alternative Polyadenylation (APA), and a subset of core clock and compensation-relevant genes are altered in ΔCFIm mutants.** Circadian bioluminescence was recorded from race tube cultures of the indicated strains grown in high amino acid race tube medium (0.17% w/v arginine, 0% glucose). Luciferase signal was acquired from the entire race tube of fungal growth (yellow lines) or from an old tissue region of the race tube (blue lines) (see S2 Movie) (standard deviation error bars). Circadian period lengths were computed for each region (Materials and methods; *N* = 4 race tubes per genotype): control: 21.7 ± 0.2 hours (whole tube), 21.4 ± 0.3 hours (old tissue); Δ*cpsf5*: 19.0 ± 0.5 hours (whole tube), 17.2 ± 0.2 hours (old tissue); Δ*cpsf6*: 19.4 ± 0.3 hours (whole tube), 17.3 ± 0.3 hours (old tissue) (**A**). Wild-type control and Δ*cpsf5* Δ*cpsf6* double mutant cellophane plate cultures were grown in constant light at 25°C for 72 hours on high nutrient medium (0.25% w/v glucose, 0.17% w/v arginine). Total RNA was extracted, and 3′ End Sequencing was performed (*N* = 2 biological replicates per genotype). Sixteen core clock and compensation genes of interest were selected, read pileup data from 3′ UTR regions were extracted, biological duplicate samples averaged together, and heatmaps were generated. Read count pileups are depicted as log₂-fold changes, and both color scales are normalized to mean counts in the wild-type dataset. Each point along the x-axis represents nucleotide coordinates from the STOP codon for each mRNA (where data from any negative/Crick strand genes are shown in reverse orientation), and genes are ordered along the y-axis by increasing 3′ UTR lengths. Asterisks (*) indicate two genes, *ckb-1* and *pabp-2*, with significantly altered APA patterns between control and mutant (**B**). Genomic tracks were generated using the Gviz package in R to visualize poly(A) read pileups in the 3′ UTR regions of *ckb-1* and *pabp-2*. In the ΔCFIm mutant, poly(A) tail locations are significantly changed for *ckb-1* and *pabp-2* (**C**). Gene expression levels of core clock and compensation genes were measured by normalizing total read counts for each gene (Materials and methods). Log₂-transformed read counts are shown. Asterisks indicate $p < 0.05$ (*) by Student *t* test comparing mutant to control levels. The *cka^prd-3^* transcript is 1.34-fold up-regulated, and the *ck-1a* transcript is 1.2-fold down-regulated in ΔCFIm (**D**). The raw data underlying parts (**A**) and (**D**) can be found in S4 Data.

transition from high-to-low amino acid levels in old tissue to reveal the shorter period defect (S2 Movie). The Δ*cpsf5* and Δ*cpsf6* mutants differ from other under-compensation mutants (Fig 3 and S3 Fig) because the short period defect was induced by amino acids, not by glucose, and because the Nutritional Compensation phenotype is specific to old tissue (Fig 4A).

## The Alternative Polyadenylation (APA) landscape is altered in Nutritional Compensation mutants

Given the large period defect observed in Δ*cpsf5* and Δ*cpsf6* mutants (Fig 4A) and other under-compensation mutants related to polyadenylation (Fig 3A), we hypothesized that poly (A) tail maintenance and concurrently the stability of a core clock mRNA(s) is required for Nutritional Compensation. To assay the biochemistry of Δ*cpsf* mutants, we required a solid medium growth regime to carefully control nutrient levels and to confidently compare results to our genetic screen for Nutritional Compensation phenotypes. *Neurospora* biochemistry is traditionally accomplished using extracts from liquid-grown cultures [57,58], where nutrient consumption rates are less well defined and less relevant to the ecological niche of the organism. A cellophane petri plate assay was developed to harvest biomolecules from solid medium cultures of *Neurospora* (Materials and methods) [34]. We confirmed that the circadian clock is equally functional on cellophane plates and in liquid cultures (S5 Fig). We next generated a Δ*cpsf5* Δ*cpsf6* double mutant strain (hereafter referred to as ΔCFIm) and confirmed that its period length and Nutritional Compensation phenotype matched results from race tube assays (S5 Fig). Biological duplicate wild-type control and ΔCFIm mutant cellophane plates were grown at 25°C under constant light for 3 days on high nutrient medium (containing 0.25% w/v glucose 0.17% w/v arginine; high-to-low nutrient transition will occur in aged tissue before 72 hours growth), and total RNA was extracted for 3′ End Sequencing (Materials and methods).

Nascent mRNA transcripts can contain multiple sites for polyadenylation to occur, which is known as Alternative Polyadenylation (APA). APA events generate mRNA isoforms with variable 3′ UTR lengths and nucleotide sequences (reviewed in [59]). Along with other factors involved in mRNA 3′ end cleavage and polyadenylation, CFIm mutants have been shown to alter APA patterns genome wide [60–62]. Changes in APA have been directly linked to RNA stability in yeast and in mammals [63,64]. We first sought to identify normal instances of APA in wild-type *Neurospora* 3′ UTRs by comparing our 3′ End Sequencing dataset to an existing 2P-Seq dataset [65] (Materials and methods). A total of 843 consensus genes (>10% of *Neurospora* 3′ UTRs) contain multiple poly(A) sites after applying a strict intersection cutoff for identification in all 4 wild-type datasets (S3 Table), which is a lower genome-wide estimate than >60% APA in mammals. Next comparing ΔCFIm and controls, we found 193 examples of APA events in controls collapsing to a single poly(A) peak in mutants (21%), 123 examples of a single poly(A) peak in controls expanding to multiple APA events in mutants (13%), and 155 examples of APA events in both control and mutant where the location of the predominant poly(A) peak was significantly changed in mutants (16%) (S4 Table and S6 Fig). Taken together, approximately 50% of the APA landscape is altered in *Neurospora* ΔCFIm mutants. Knockdown of the mammalian CFIm complex causes global 3′ UTR shortening, as proximal poly(A) sites become preferred over distal poly(A) sites [60–62]. In *Neurospora*, we find that a distal-to-proximal shift occurs in a majority (63%) of the 155 altered APA events in the ΔCFIm mutant.

Nine core clock and compensation genes are among the consensus list of 3′ UTRs with APA events: *ck-1a* (NCU00685), *frq* (NCU02265), *upf1*[prd-6] (NCU04242), *ckb-1* (*NCU05485*), *rco-1* (NCU06205), *pkac-1* (NCU06240), NCU06565, *prd-1* (NCU07839), and *nab2*

(NCU16397) (S3 Table). Given the strict cutoff for consensus APA, we added 7 additional genes after visual inspection: *set-2* (NCU00269), *wc-2* (NCU00902), *set-1* (NCU01206), *csp-1* (NCU02713), *cka^{prd-3}* (NCU03124), *frh* (NCU03363), and *pabp-2* (NCU03946). 3′ UTR regions containing APA events were visualized in a heatmap (Fig 4B). Two out of 16 genes of interest, *ckb-1* and *pabp-2*, are among the list of genes with altered APA patterns in the ΔCFIm mutant (S4 Table and Fig 4B, asterisks *). poly(A) read pileups were visualized for 3′ UTRs of *ckb-1* and *pabp-2* to confirm significant reorganization of poly(A) tail locations in the ΔCFIm mutant (Fig 4C). Following our original hypothesis, altered 3′ UTR length in ΔCFIm should lead to altered mRNA stability and changes in gene expression compared to control samples. An extremely slight and statistically nonsignificant increase (*t* test, *p* = 0.5) was observed for both *ckb-1* and *pabp-2* gene expression levels (Fig 4D). *ckb-1* encodes the regulatory subunit of *Neurospora* CKII, and interestingly, the catalytic alpha subunit of CKII (*cka^{prd-3}*) increased significantly in the ΔCFIm mutant, despite no visible changes in 3′ UTR poly(A) sequence coverage for *cka^{prd-3}* (Fig 4B). In addition, *ck-1a* gene expression levels are modestly (1.2-fold) but significantly decreased in ΔCFIm (Fig 4D), a result that is counterintuitive given the mutant's short period length phenotype [52]. Future work will determine whether overexpression of CKII underlies the ΔCFIm short period length and Nutritional Compensation phenotypes.

## Chromatin modifiers emerge as key regulators of circadian period length across nutrients

We next examined the 4 new over-compensation mutants identified in our genetic screen, where period lengthens as a function of nutrient levels (Fig 5A and S1 Table). The wild-type *Neurospora* clock is slightly under-compensated (Figs 1–3), and, therefore, this group of over-compensation mutants, together with *prd-1* (Fig 2B) [37], represent clear and bona fide Nutritional Compensation defects. Protein Kinase A (*pkac-1*) shows extended compensation, or approximately the same period length at zero and high nutrients (Fig 5A). The effect of loss of PKA on Nutritional Compensation is subtle and likely due to its phosphorylation and inhibition of RCM-1, activating WCC-independent *frq* transcription—RCM-1 normally acts as a general transcription corepressor with RCO-1 and prevents *frq* transcription in a Δ*wc-1* or Δ*wc-2* background [35,66]; however, a Δ*rcm-1* mutant did have normal Nutritional Compensation in our screen (S1 Table). PKA also directly regulates WCC and FRQ by phosphorylation [67]; thus, an alternative explanation for its extended compensation phenotype (and the lack of phenotype in Δ*rcm-1*) could be nutrient-dependent posttranslational regulation of core clock proteins.

RBG-28 (NCU02961, FGSC16412) is a broadly conserved ribosome biogenesis factor in fungi, currently uncharacterized in the *Neurospora* literature but homologous to *RNH70* (YGR276C) in *S. cerevisiae*. Rnh70p (Rex1p) is a 3′- to 5′ exonuclease involved in 5S and 5.8S rRNA processing [68]. The completely arrhythmic phenotype of Δ*rbg-28* on high nutrients (Fig 5A) is intriguing and implicates translational machinery in Nutritional Compensation for the first time (see Discussion).

Half of the novel over-compensation mutants are involved in chromatin regulation and both contain SET domains (reviewed in [69]). SET-1 (NCU01206) is a histone H3 lysine 4 (K4) methyltransferase and the catalytic subunit of the COMPASS complex in *Neurospora*. Δ*set-1* was previously reported to be arrhythmic [70], but here we find an increasing period length as a function of glucose levels (Fig 5A). SET-1 was convincingly shown to regulate methylation levels and transcriptional repression at the *frq* locus [70,71], and upon reanalysis of the Δ*set-1* transcriptome, *frq* is the only core clock gene significantly altered in the Δ*set-1* mutant due to loss of repression (Fig 5B). The Δ*set-1* nutritional over-compensation

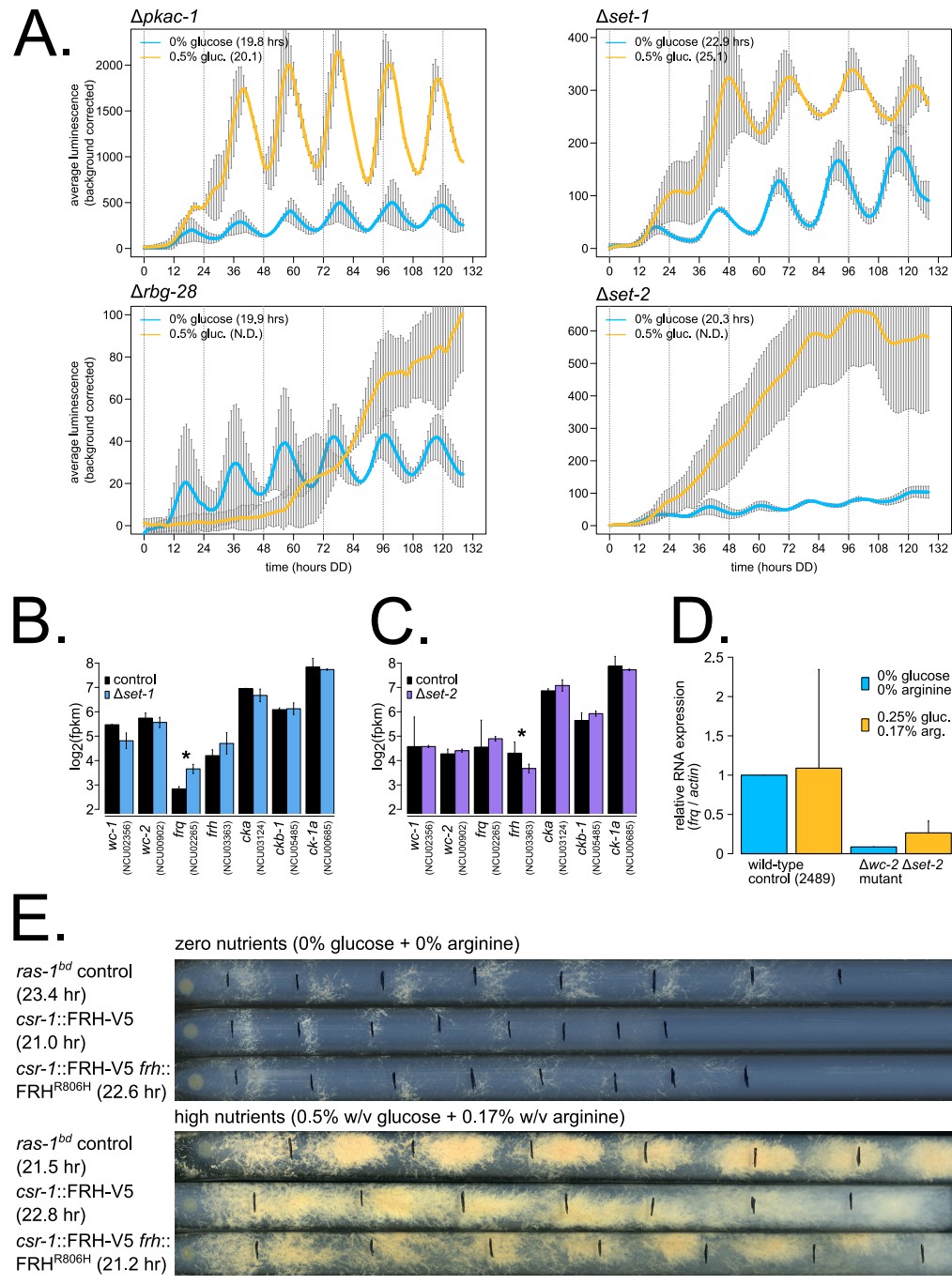

**Fig 5. Nutritional over-compensation mutants are enriched for chromatin regulators, implicating *frh* regulation in Nutritional Compensation mechanism.** Circadian bioluminescence was recorded from race tube cultures of the indicated deletion mutants. High nutrient medium (yellow lines) contained 0.5% w/v glucose 0.17% w/v arginine, and zero nutrient medium (blue lines) contained 0% glucose 0% arginine (standard deviation error bars). Period lengths are indicated as insets ($N = 2$–$4$ biological replicates per nutrient concentration) (**A**). RNA-Sequencing data were mined from a previous study [71] (Materials and methods), where 2% (high) glucose liquid cultures were harvested at circadian time point DD24. Log$_2$-transformed FPKM values are shown for core clock gene expression levels ($N = 2$ biological replicates per genotype). The asterisk indicates $p = 0.05$ (*) by Student $t$ test comparing mutant to control levels. The *frq* transcript is 1.78-fold more abundant in Δ*set-1* (**B**). RNA-Sequencing data were mined from a previous study [73] (Materials and methods), where 1.5% (high) sucrose liquid cultures were sampled. Log$_2$-transformed FPKM values are shown for core clock gene expression levels ($N = 2$ biological replicates for Δ*set-2*, and $N = 6$ biological replicates for controls). The asterisk indicates $p < 0.05$ (*) by Student $t$ test comparing mutant to control levels. The *frh* transcript is 1.61-fold less abundant in Δ*set-2* (**C**). RNA expression levels were measured from the indicated genotypes

grown on 0.25% w/v glucose + 0.17% w/v arginine high nutrients versus zero nutrient (0% glucose + 0% arginine) solid medium for 72 hours in constant light at 25˚C ($N$ = 2 biological replicate samples; $N$ = 3 technical replicates for each RT-qPCR reaction; standard deviation error bars). *frq* mRNA expression levels are detectable in the WCC-independent transcription mutant Δ*wc-2* Δ*set-2* at zero nutrients but are reduced compared to the high nutrient condition, suggesting that nutrients contribute to the extent of WCC-independent *frq* transcription (**D**). The entire *frh* locus (plus 1 kb of its upstream promoter sequence, a C-terminal V5 epitope tag, and 500 bp of its 3′ UTR) was integrated at the *csr-1* locus (strain # 1192–1). In a second mutant with full-length *frh* integrated at *csr-1*, the endogenous *frh* locus was replaced with the FRH[R806H] mutant and marked with ignite resistance (strain # 1163–1). The *frh* + *frh* overexpression mutant, the *frh* + *frh*[R806H] mutant, and control strain were compared by race tube ($N$ = 10–19 biological replicates). The *frh* overexpression condition led to nutritional over-compensation (zero nutrients: 21.0 ± 1.3 hour [SEM: 0.29; $N$ = 19]; high nutrients: 22.8 ± 0.8 hour [SEM: 0.19; $N$ = 19]). The *frh* + *frh*[R806H] mutant (zero nutrients: 22.6 ± 1.3 hour [SEM: 0.42; $N$ = 10]; high nutrients: 21.2 ± 1.1 hour [SEM: 0.35; $N$ = 10]) and control (zero nutrients: 23.4 ± 0.5 hour [SEM: 0.13; $N$ = 13]; high nutrients: 21.5 ± 0.4 hour [SEM: 0.13; $N$ = 12]) showed slight under-compensation to nutrients (**E**). The raw data underlying parts (**A**–**E**) can be found in S5 Data.

phenotype is consistent with loss of chromatin repression on *frq*—as glucose levels increase, more cellular energy is available for transcription/translation, and circadian period is lengthened due to high *frq* levels and prolonged negative feedback. This SET-1 nutritional mechanism is analogous to CKII's role in regulating the larger pool of FRQ protein at higher temperatures in Temperature Compensation [21].

Because the Δ*set-1* mutant was reported to be arrhythmic in previous work [70], yet we identified period lengthening as a function of nutrients, we investigated whether the role of SET-1 in Nutritional Compensation involved its reported role in the COMPASS complex. The *frq* clock box transcriptional reporter was integrated into individual knockouts of COMPASS complex subunits, which were previously assayed only by race tube [70]. Consistent with previous findings, *swd2* (NCU07885) and *sdc1* (NCU03037) are not required for circadian function and have a wild-type period length by luciferase assay (S7 Fig) and by race tube [70]. Curiously, Δ*swd1* (NCU02104) had an intact clock by luciferase (S7 Fig) despite its arrhythmic phenotype by race tube [70]. *set-1* and *swd3* (NCU03244) mutants each had approximately 24-hour long period phenotypes by luciferase assay (S7 Fig). To investigate whether the long period phenotype of Δ*set-1* was exclusively due to loss of function of the COMPASS complex, we generated a double mutant Δ*set-1* Δ*swd3* and assayed period length and Nutritional Compensation. The single and double mutants each showed equivalent period lengthening and extended or over-compensation to nutrients (S7 Fig). Thus, SET-1 and the COMPASS complex are required for functional Nutritional Compensation through direct regulation of *frq* mRNA abundance.

SET-2 (NCU00269) is a histone H3 K36 methyltransferase in *Neurospora*, which deposits inhibitory chromatin marks in actively transcribed regions via a physical association with RNA Polymerase II and prevents improper transcription initiation inside coding regions [72,73]. Δ*set-2* was also previously reported to be arrhythmic [35,74], but here we find low amplitude Δ*set-2* rhythms that are completely intact with a short period length on nutrient starvation medium (Fig 5A) and an arrhythmic phenotype observed at high nutrients (S3 Movie). SET-2 is required to maintain H3K36me2 and H3K36me3 marks across the *frq* locus, and the Δ*set-2* mutant results in hyper-acetylation of *frq*, improper activation of WCC-independent *frq* transcription, constitutively high *frq* expression levels, and, presumably, the arrhythmic clock phenotype observed under high nutrients [35,74]. Our result indicates for the first time that WCC-independent *frq* transcription is nutrient dependent and only reaches levels sufficient for arrhythmicity at high nutrient levels. To test this hypothesis, we assayed *frq* mRNA expression level by quantitative reverse transcription PCR (RT-qPCR) on high versus zero nutrients in control compared to a Δ*wc-2* Δ*set-2* double mutant. We observed less WCC-

independent *frq* transcription in the zero nutrient condition compared to high nutrients under constant light conditions (Fig 5D).

However, low WCC-independent *frq* transcription does not explain the short period length of the Δ*set-2* mutant in zero nutrient medium (Fig 5A), as low WCC expression results in lengthened periods [75]. Curiously, upon mining existing data for the Δ*set-2* transcriptome, *frq* levels were not significantly increased, and instead *frh* levels were down-regulated (Fig 5C). *frh* is among the top down-regulated genes during glucose starvation (S4 Fig), and perhaps Δ*set-2* further affects *frh* transcription, leading to a circadian period change. Notably, two RNA helicases physically associated with the mammalian negative arm complex, DDX5 and DHX9, show a short period length upon siRNA knockdown [76], and so decreased levels of the *frh* helicase in Δ*set-2* could potentially explain its short period phenotype. To test whether *frh* gene expression regulation is important for Nutritional Compensation, we used a mutant background with a second copy of full-length *frh* expressed at the *csr-1* locus (strain # 1192–1 from [54]). Increased expression of *frh* directly caused a nutritional over-compensation phenotype, demonstrating that down-regulation of *frh* mRNA under starvation conditions is physiologically relevant to the circadian clock (Fig 5E and S4 Fig). Furthermore, the extra copy of FRH requires circadian functionality to produce the over-compensation phenotype. The FRH[R806H] mutant, which preserves its helicase activity but cannot bind FRQ [77], expressed as the second copy of FRH (strain # 1163–1 from [54]) matches the wild-type nutritional under-compensation profile (Fig 5E).

Taken together, our genetic screen revealed mRNA stability, polyadenylation, and chromatin modifier pathways converging on gene expression regulation to enact circadian Nutritional Compensation in fungi. Nutritional Compensation effectors then directly regulate core clock factors, such as *wc-1*, *wc-2*, *frh*, and *ck-1a* gene expression levels, to maintain circadian period length across nutrient environments.

## Nutritional Compensation also functions in a mammalian circadian system

Compensation is a defining principle of circadian oscillators. Temperature Compensation mechanisms are active in mammalian tissue culture [17,18], despite long evolutionary time-scales in homeothermic organisms. Thus, we hypothesized that Nutritional Compensation is also a conserved feature between the fungal and mammalian circadian clocks. Four previous studies hinted that Nutritional Compensation mechanisms may be actively maintaining circadian period length across physiologically relevant nutrient environments.

Inhibiting transcription using α-amanitin or actinomycin D against RNA Polymerase II activity led to dose-dependent shortening of the NIH3T3 period length [78]. In other words, the mammalian circadian oscillator is over-compensated with respect to transcription rates, reminiscent of temperature over-compensation observed in *Gonyaulax* [2]. Similarly, induction of autophagy and amino acid starvation shortens period length in mouse embryonic fibroblast (MEFs) [79], once again indicating an over-compensation phenotype to amino acid levels in mammals. Varying levels of FBS do not alter the circadian period length in NIH3T3 cells [80] despite substantial changes in cell growth rate. Rhythms at the single-cell level of MEFs grown in microfluidic devices show slight period shortening when cells were given fresh medium every hour compared to a single medium supply at the beginning of the experiment, suggesting instead nutritional under-compensation [81]. Thus, preliminary evidence is consistent with functional Nutritional Compensation in the mammalian circadian system.

We used a U2OS *Bmal1-dLuc* reporter cell line to compare rhythms in high versus low glucose medium and found a slight under-compensation phenotype (S8 Fig), similar to the single-cell rhythms report [81]. Period length was 0.5 hours shorter in 25 mM (high) glucose

compared to 5.56 mM (low) glucose, trending in the same direction as the approximately 1.5-hour period shortening seen for the wild-type *Neurospora* clock across glucose concentrations (Figs 1–3). In this circadian assay, *Bmal1-dLuc* cells have reached confluency and are no longer actively dividing, but metabolism and drug responses of cultured cells are reported to be significantly different in high versus low glucose tissue culture medium (reviewed in [82]).

Maintenance of circadian period length between high and low glucose medium and a small amount of literature precedent does not, however, prove the relevance of Nutritional Compensation to the mammalian clock. A mutant phenotype showing nutrient-dependent period changes (i.e., defective Nutritional Compensation) would provide much stronger evidence for the biological relevance of compensation. Therefore, we selected two compensation mutants from our fungal genetic screen—Δ*cpsf6* (NCU02152) and Δ*set-2* (NCU00269)—and identified the homologous human genes *CPSF6* and *SETD2* (reciprocal BLAST e-values = $7e^{-6}/7e^{-8}$ for Cpsf6 and $4.4e^{-61}/5.7e^{-62}$ for Set-2). If the circadian functions of *CPSF6* or *SETD2* are indeed conserved with *Neurospora*, we expected to observe a period length change as well as a Nutritional Compensation defect. *CPSF6* and *SETD2* were not among the hits from a genome-wide screen for period length defects using U2OS cells [83] or tested in a kinase/phosphatase siRNA screen [84]; however, visual inspection of the genome-wide screen data confirmed that a subset of the pooled siRNAs did show period effects after *CPSF6* or *SETD2* knockdown (source: BioGPS database, "Circadian Genomics Screen" plugin). Thus, we used the U2OS siRNA genome-wide screen database [83] as preliminary data to select *CPSF6* and *SETD2* for further study.

*CPSF6* and *SETD2* were knocked down using siRNAs in *Bmal1-dLuc* cells, and period length was measured from high and low glucose medium (Fig 6). AllStars Negative Control siRNA, which does not target any known mammalian transcript, was used as an internal control for each biological replicate assay, and *CRY2* siRNA knockdown was used as a positive control for a known long period phenotype [85,86]. Control *Bmal1-dLuc* cells had a slight under-compensation phenotype (Fig 6A), matching preliminary results (S8 Fig). *CRY2* knockdown lengthened period by approximately 4 hours in both high and low glucose conditions, indicating no effect on Nutritional Compensation (Fig 6A). *CPSF6* knockdown lengthened period by approximately 1.5 hours in high glucose and further lengthened period by approximately 3 hours in low glucose, indicating an under-compensation phenotype compared to control cells (Fig 6A). The long period observed in *CPSF6* knockdowns was the opposite of the short period phenotype in *Neurospora* (Fig 4A). Most interestingly, *SETD2* knockdown drastically increased the amplitude and magnitude of the *Bmal1-dLuc* transcriptional reporter compared to controls. As with *Neurospora*, *SETD2* knockdown rhythms were less robust (Figs 5A and 6A). *SETD2* knockdown lengthened period by approximately 2.5 hours in high glucose but only lengthened period by approximately 1 hour in low glucose (Fig 6A). To further validate nutritional over-compensation in *SETD2* knockdowns, *Bmal1-dLuc* cells were compared across higher concentrations of glucose with more robust rhythms (Fig 6B). Just like *Neurospora*, *SETD2* mutants show a nutritional over-compensation phenotype. This genetic evidence strongly suggests that Nutritional Compensation mechanisms also regulate the mammalian circadian clock in physiologically relevant environments.

## Discussion

We present the largest genetic screen to date for mutants displaying altered circadian Nutritional Compensation (S1 and S2 Tables and Fig 2C). Together with a recent survey of 177 transcription factor knockouts [50], circadian functional genomics is well underway utilizing the *Neurospora* deletion collection [43]. In this study, 16 new Nutritional Compensation mutants

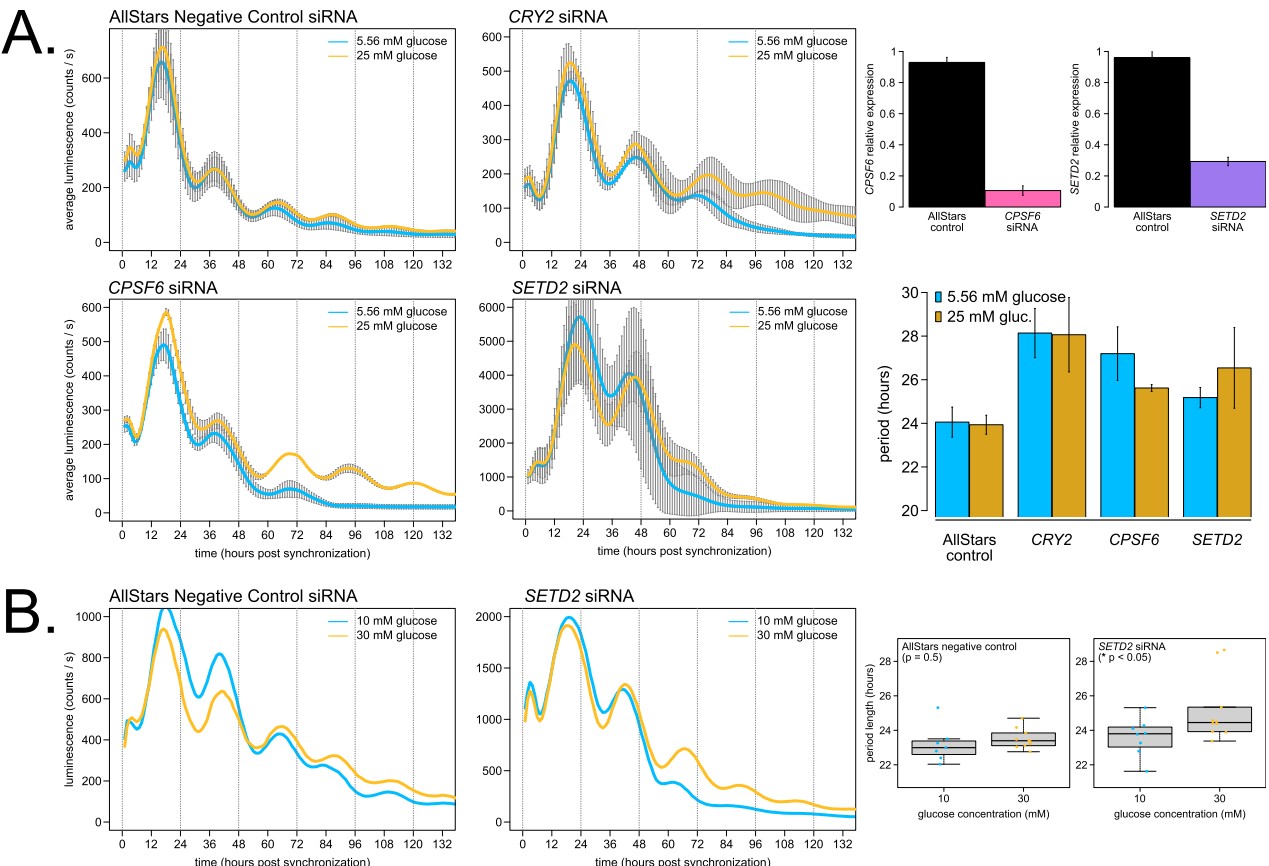

**Fig 6. *CPSF6* and *SETD2* are required for normal Nutritional Compensation of U2OS *Bmal1-dLuc* cells cultured in high versus low glucose.**
U2OS cells were transfected with the indicated siRNAs (15 pmol total), incubated for 2 days, and *Bmal1-dLuc* rhythms were measured for 5–6 days following dexamethasone synchronization. Knocked-down cells were assayed in either MEM high glucose (yellow lines) or MEM low glucose (blue lines) medium. Averaged luciferase traces are shown for each glucose concentration without detrending or further data manipulation ($N = 6$ biological/technical replicates for AllStars negative controls; $N = 4$ biological/technical replicates for other siRNA knockdowns per nutrient concentration; standard deviation error bars). siRNA knockdown efficiency was measured using RT-qPCR relative to nontransfected control samples (Materials and methods). Approximately 90% knockdown was achieved for *CPSF6*, and approximately 70% knockdown for *SETD2* under these experimental conditions ($N = 2$ biological replicate samples each with $N = 3$ technical replicate RT-qPCR reactions). Period lengths were calculated and averaged for all replicates (**A**). siRNA knocked-down cells were assayed in either DMEM 30 mM glucose (yellow lines) or DMEM 10 mM glucose (blue lines) medium. One representative luciferase trace is shown ($N = 7$–10 biological/technical replicates per nutrient concentration). Period lengths were calculated and averaged for all replicates and plotted as boxplots (**B**). Average period lengths were as follows: $23.2 \pm 1.1$ hours (low glucose control) and $23.5 \pm 0.6$ hours (high glucose control); $23.6 \pm 1.1$ hours (low glucose *SETD2* knockdown); and $25.2 \pm 1.8$ hours (high glucose *SETD2* knockdown) (* $p < 0.05$, Student *t* test). The raw data underlying parts (**A**) and (**B**) can be found in S6 Data.

were identified and characterized along with the 4 previously characterized mutants (Δ*csp-1*, *prd-1*, Δ*rco-1*, and Δ*ras2*)—fungal Nutritional Compensation occurs at the level of gene regulation and involves transcription factors, RNA helicases, chromatin modifiers, and polyadenylation machinery. Nutritional Compensation effectors are responsible for directly regulating the core oscillator to maintain the circadian period length across different nutrient levels. In high nutrient environments, CSP-1 forms a negative feedback loop on WCC expression and activity by inhibiting *wc-1* overexpression and preventing nutritional under-compensation [32,33]. RCO-1, SET-2, and possibly PKA are required to block WCC-independent *frq* transcription in high nutrient conditions (Fig 5) [34,35,74]. More broadly, repressive chromatin marks at the *frq* locus, including those regulated by SET-2, SET-1, and the COMPASS complex, are required to maintain Nutritional Compensation, especially in high nutrients (Fig 5 and S7

Fig). The NMD machinery is required for circadian function through its regulation of *ck-1a* expression [52], and *ck-1a* transcript stability appears to undergo additional, nutrient-dependent regulation because deletion of the *ck-1a* 3′ UTR (its NMD-targeting sequence) causes nutritional under-compensation (Fig 3C). The APA landscape in *Neurospora* is dynamic across nutrient conditions and requires CFIm complex activity (Fig 4).

This screen and associated candidates have opened up several avenues for further work on Nutritional Compensation mechanisms in *Neurospora*. For instance, PRD-1 RNA helicase target genes under high glucose conditions are completely unknown. In this study, we provide some evidence that CKII activity is altered in ΔCFIm mutants (Fig 4), but the effect of CKII overexpression on both circadian period length and Nutritional Compensation remains to be tested. Genetic screens extending from this work can examine both transcription factor and chromatin modifier knockouts for Nutritional Compensation phenotypes. Curiously, the subset of chromatin readers and writers that have been screened to date (using standard medium conditions) have an unusually high approximately 15% hit rate for circadian period defects [87,88]. In this study, chromatin modifiers not only regulate chromatin state at the *frq* locus, but dynamic expression of *frh* across nutrient levels may also be involved in compensation, potentially mediated by SET-2 (Fig 5C and 5E). Intriguingly, quantitative modeling has suggested that nutrient-dependent FRH sequestration away from the negative arm complex is a plausible mechanism for Nutritional Compensation [89]. Mechanistic work in prokaryotes has indicated that Nutritional Compensation can be derived from the same core clock enzyme, KaiC, through its protein domains with equal and opposite balancing reaction rates [90,91]. In fungi and perhaps other eukaryotic clocks, Nutritional Compensation appears to be maintained through regulation of multiple different core clock factors, including *wc-1*, *wc-2*, *frh*, and *ck-1a*. Nutritional Compensation mechanisms very likely extend to other physiological environmental variables such as nitrogen sources and levels [92], vitamins, and soil pH [93].

This work establishes Nutritional Compensation in the mammalian circadian clock for the first time, and its clinical relevance may be high. There is a direct link between pancreatic clock function and risk for type 2 diabetes [94]. Fasting glucose concentration in human serum is approximately 5 mM in healthy controls, but glucose levels can increase by 3+ orders of magnitude in type 2 diabetic hyperglycemia [95,96]. Thus, deleterious SNPs in Nutritional Compensation-relevant genes could exacerbate disease outcomes when cellular clocks encounter nutrients outside of the physiological range. In addition to Nutritional Compensation, there is a large body of quality literature describing the set of metabolites and metabolic enzymes that can directly feed back and affect circadian function, including adenosine monophosphate (AMP) and AMP kinase, nicotinamide adenine dinucleotide (NAD+) and SIRT1 activity, acetylglucosamine (O-GlcNAc) [97], and mTOR activity [98] (reviewed in [99–102]). In *Neurospora*, extensive metabolic rhythms are also present [45,103], and rhythmic metabolic reaction fluxes likely function similarly to the mammalian clock [104–106]. For the mammalian clock, it will be critical to define "physiologically relevant" nutrient levels, which could vary by organ or even cell type. A high-fat diet has been shown to lengthen circadian behavioral rhythms by approximately 0.5 hours [107], suggesting an over-compensation phenotype. In this study, we observed nutritional under-compensation in U2OS osteosarcoma cells (Fig 6). Does a high-fat diet push cells out of the physiological range of compensation for the circadian oscillator, or are different mammalian cell types differentially over- or under-compensated to nutrients? Notably, time-restricted feeding has been shown to alleviate circadian disruption and some metabolic consequences of a high-fat diet [108,109]. Future work will implement more metabolically relevant cell types, such as hepatocytes and adipocytes [110], to answer these questions about mammalian clock compensation.

Circadian control of polyadenylation is an emerging topic. Modern transcriptomic approaches have identified hundreds of rhythmic APA events in mice and in plants [111–114]. We and others have begun to define the set of APA events in *Neurospora* [65] (S3 Table), which can be extended to circadian APA events. In *Neurospora*, CPSF5 may physically interact with the negative arm complex [56]. Two cleavage and polyadenylation factors (CPSF1 and 7) were found to physically interact with the negative arm complex in mice [115], and *CPSF6* oscillates at the transcriptional level in mouse kidney and brain [116]. During preparation of this manuscript, circadian colleagues reported the long period length of *CPSF6* knockdown [117], which our results further support (Fig 6A). Interestingly, *CPSF6* knockdown cells showed a temperature under-compensation defect, and multiomics identified *EIF2S1* as the key effector gene upon *CPSF6* knockdown [117]. The *Neurospora* homolog of *EIF2S1* is eIF2α/NCU08277 (reciprocal BLAST e-values = $6e^{-91}/1e^{-81}$), which is a central hub of rhythmic translation initiation peaking in the subjective evening [118]. We identified a ribosome biogenesis exonuclease, RBG-28, which is required for rhythmicity under high nutrient conditions (Fig 5A). Transcription, RNA processing reactions (such as NMD, splicing, and polyadenylation) and translation are tightly coupled processes. In fact, rhythmic polyadenylation of rRNAs has been linked to translational rhythms in mouse liver [119]. Nutritional Compensation pathways likely occur at multiple steps in the gene expression of multiple core clock components.

By suggesting *CPSF6* and *SETD2* as targets, *Neurospora* functional genetics has again informed mammalian-relevant circadian mechanisms (reviewed in [120]), and the over-compensation defect of *SETD2* provides affirming evidence of eukaryotic Nutritional Compensation outside of *Neurospora* (Fig 6). A handful of previous studies have implicated histone methyltransferases in mammalian circadian function, including both activating [121,122] and repressive chromatin marks [123]. *SETD2* joins a growing list of circadianly relevant histone methyltransferases and chromatin modifiers. In fact, recent work has demonstrated a novel function for a key histone methyltransferase in the circadian TTFL (TRITHORAX in insect, MLL1 in mammals), further highlighting the utility of circadian model systems for understanding the mammalian clock [124].

## Materials and methods

### *Neurospora* strains, growth conditions, and genetic screen

Strains used in this study were derived from the wild-type background (FGSC2489 *mat* A), *ras-1^bd* background (87–3 *mat* a or 328–4 *mat* A), or the Fungal Genetics Stock Center (FGSC) knockout collection as indicated (S1 Table). Strains were constructed by transformation or by sexual crosses using standard *Neurospora* methods (http://www.fgsc.net/Neurospora/NeurosporaProtocolGuide.htm). The *frq* clock box transcriptional reporter was transformed and used as previously described [52]. The fungal biomass reporter gene (S1 Fig) is composed of 430 bp of the *gpd* promoter from *Cochliobolus heterostrophus* driving constitutive levels of codon optimized *luciferase* and integrated at the *csr-1* (NCU00726) locus [125].

All race tubes contain a base medium of 1X Vogel's Salts, 1.5% w/v Noble agar (Thermo Fisher # J10907), and 50 ng/ml biotin. Noble agar was used instead of standard bacteriological agar (Thermo Fisher # J10906) because impurities in bacteriological agar can be metabolized by *Neurospora* and interfere with accurate quantification of Nutritional Compensation phenotypes [37]. Glucose and arginine were supplemented into race tube medium as indicated. High glucose was defined as 0.5% w/v (27.8 mM) based on literature precedent [32,34] and based on growth rate and period length similarity to higher glucose concentrations (Fig 1A and S1G Fig). High arginine was defined as 0.17% w/v because concentrations higher than 0.5% w/v

interfere with circadian banding in race tube assays [5]. To optimally visualize and quantify the circadian banding pattern in the primary genetic screen (Fig 2C), screen medium contained 1X Vogel's Salts, 1.5% Noble agar, 50 ng/ml biotin, and 0.17% arginine. To quantify period lengths in carbon and nitrogen starvation conditions, the secondary screen medium contained 1X Vogel's Salts, 1.5% Noble agar, 50 ng/ml biotin, and 25 µM luciferin (GoldBio # 115144-35-9). The tertiary screen medium contained 1X Vogel's Salts, 1.5% Noble agar, 50 ng/ml biotin, 25 µM luciferin, and glucose/arginine levels as indicated (Figs 3–5). For Temperature Compensation 96-well plate experiments (Fig 2A) and COMPASS complex subunit knockout assays (S7A and S7B Fig), the standard medium recipe contained 1X Vogel's Salts, 1.5% bacteriological agar, 50 ng/ml biotin, 25 µM luciferin, 0.03% w/v glucose, and 0.05% w/v arginine.

Liquid medium cultures were grown from fungal plugs in Bird Medium + 1.8% w/v glucose (S5 Fig) as previously described [58]. Solid medium cultures were implemented to determine mRNA (or protein) levels from Nutritional Compensation mutants of interest with modifications to previous work [34]. Medium was poured into 100-mm petri plates and cooled to solidify (approximately 20 ml per plate; 1X Vogel's Salts, 1.5% Noble agar, 50 ng/ml biotin, 25 µM luciferin, and glucose/arginine levels as indicated). Cellophane (Idea Scientific # 1080) paper discs were cut to the size of 100-mm plates and autoclaved to sterilize. A sterile cellophane disc was then placed on top of the solidified medium. Conidia from strains of interest were resuspended in 100 µl of sterile water, vortexed to mix, pipetted on top of the cellophane disc, and spread with a sterile plate spreader (in order to maintain approximately the same conidial age across the cellophane plate). Inoculated plates were then covered with a Breathe-Easy strip for gas exchange (USA Scientific # 9123–6100). After growing tissue on cellophane plates for the indicated amount of time, mycelia and conidia were harvested from atop the cellophane layer by scraping with a 1,000-µl pipette tip. Harvested fungal tissue (approximately the size of 1 US quarter per each 100-mm cellophane plate) was rapidly hand dried using paper towels and an Eppendorf tube rack and flash frozen in liquid nitrogen for storage before biochemical extraction.

Most strains were genotyped by growth on selective medium (5 µg/ml cyclosporine A and/or 200 to 300 µg/ml Hygromycin). Key strains were genotyped by PCR as previously described [52] using genotyping primers:

$ck$-$1a^{LONG}$-$VHF\Delta3'UTR$::hyg$^R$ (NCU00685 $\Delta3'$UTR): 5′ GCTGCTGCTCGTAAGGAC 3′ and 5′ CATCAGCTCATCGAGAGCCTG 3′

$\Delta cpsf5$::hyg$^R$ (FGSC KO mutant): 5′ CTCTGGTCGAGAACACTGCG 3′ and 5′ CAGGCTCTCGATGAGCTGATG 3′

$\Delta cpsf6$::hyg$^R$ (FGSC KO mutant): 5′ CACCAACCCTAACCCGTGAT 3′ and 5′ CAGGCTCTCGATGAGCTGATG 3′

$\Delta set$-$2$:hyg$^R$ (FGSC KO mutant): 5′ GACGTCATCGGTGTTGAGAC 3′ and 5′ CAGGCTCTC GATGAGCTGATG 3′

$\Delta wc$-$2$:hyg$^R$ (FGSC11124 KO mutant): 5′ GGTGCTATGTACTCTGGGAT 3′ and 5′ CAGGCTCTCGATGAGCTGATG 3′

$\Delta set$-$1$:hyg$^R$ (FGSC KO mutant): 5′ AGTGGAAAGTCGAGCATTTGC 3′ and 5′ CAGGCTCTCGATGAGCTGATG 3′

$\Delta swd1$:hyg$^R$ (FGSC KO mutant): 5′ TGTAGTCGCGCAGGTTCTGG 3′ and 5′ CAGGCTCTCGATGAGCTGATG 3′

Δ*swd2*:hyg^R (FGSC KO mutant): 5′ TCAGGGAGAACTCAAGTCGAGT 3′ and 5′ CAGGCTCTCGATGAGCTGATG 3′

Δ*swd3*:hyg^R (FGSC KO mutant): 5′ AGTTTGGAGATCTCCCAGGACTG 3′ and 5′ CAGGCTCTCGATGAGCTGATG 3′

Δ*sdc1*:hyg^R (FGSC KO mutant): 5′ AGAGCAAATGTGCAGCTTGCT 3′ and 5′ CAGGCTCTCGATGAGCTGATG 3′

### *Neurospora* luciferase reporter detection and data analysis

The 96-well plates were inoculated with conidial suspensions and entrained in 12-hour light: dark cycles for 2 days in a Percival incubator at 25°C. Temperature inside the Percival incubator was monitored using a HOBO logger device (Onset # MX2202) during entrainment and free run. Race tubes were entrained in constant light at 25°C for 3 to 24 hours (mean entrainment time for all experiments was 14 hours in LL/overnight). Entrained 96-well plates or race tubes were then transferred into constant darkness to initiate the circadian free run. Individual race tubes were separated by approximately 3 cm tall strips of 6-ply black railroad board paper to prevent contamination of light signal between cultures. Luminescence was recorded using a Pixis 1024B CCD camera (Princeton Instruments). Bioluminescent signal was acquired for 10 to 15 minutes every hour using LightField software (Princeton Instruments, 64-bit version 6.10.1).

The average bioluminescent intensity of each 96-well or race tube was determined using a custom ImageJ Macro with background correction for each image [51,126]. Most race tube period lengths reported in this study were derived from luciferase signal measurements across the entire race tube (Figs 1, 3 and 5). However, the long period defect in the *prd-1* mutant only occurs at the growth front (i.e., high nutrients) region of fungal tissue [37]. For the *prd-1* strain, an ImageJ macro was modified to quantify only the fungal growth front (Fig 2B). On the other hand, the Δ*cpsf5* and Δ*cpsf6* mutants showed additional period shortening in aged tissue (Fig 4A) (S2 Movie). For the *cpsf* mutants, an ImageJ macro was modified to quantify only the old tissue region. Custom ImageJ Macros to quantify the growth front or old tissue regions of race tubes from Princeton *.spe image files are available at: https://github.com/cmk35.

To calculate the circadian period length, background-corrected luminescence traces were run through two different algorithms and averaged as previously described [52]. Race tubes period lengths were measured using ChronOSX 2.1 software. For Temperature Compensation experiments, the $Q_{10}$ temperature coefficient was calculated using the formula: $[(\text{frequency of clock at } 30°C) / (\text{frequency of clock at } 20°C)]^{[10°C / (30°C- 20°C)]}$, where frequency = period length$^{-1}$.

### *Neurospora* RNA isolation and 3′ End Sequencing analyses

Frozen *Neurospora* tissue was ground in liquid nitrogen with a mortar and pestle. Total RNA was extracted with TRIzol (Invitrogen # 15596026) and the Direct-zol RNA MicroPrep kit (Zymo Research # R2060) according to the manufacturer's instructions and including the on-column DNAse I treatment step (Roche # 04 716 728 001, 10 U/µl stock, 30 U used per sample). Total RNA samples were prepared for Northern Blotting, RT-qPCR, 3′ End Sequencing, or stored at −80°C.

Northern blotting was performed as previously described [52] with slight modifications. Equal amounts of total RNA (7 µg) were loaded per lane of a 0.8% w/v agarose gel (S5 Fig). For blot visualization, anti-Digoxigenin-AP Fab fragments was purchased from Sigma (Roche # 11 093 274 910) and used at 1:10,000 (75 mU/ml).

*Neurospora* RT-qPCR was performed as previously described [52]. cDNA was synthesized using the SuperScript IV First-Strand synthesis kit (Invitrogen # 18091050). RT-qPCR was performed using SYBR green master mix (Qiagen # 204054) and a StepOne Plus Real-Time PCR System (Applied Biosystems). $C_t$ values were determined using StepOne software (Life Technologies) and normalized to the *actin* gene ($\Delta C_t$). The $\Delta\Delta C_t$ method was used to determine mRNA levels relative to the zero nutrient control samples. Relevant RT-qPCR primer sequences are as follows: *frq* (NCU02265): 5′ TGGCTCGGATAAGAATGGTC 3′ and 5′ ATGAAAGGTGTCCGAAGGTG 3′. *actin* (NCU04173): 5′ GTCCCCGTCATCATGGTATC 3′ and 5′ CTTCTCCATGTCGTCCCAGT 3′.

For 3′ End Sequencing (Fig 4 and S6 Fig), RNA was extracted from solid medium high nutrient cultures (0.25% w/v glucose + 0.17% w/v arginine) grown for 72 hours at 25˚C. Total RNA was submitted to the Dartmouth Genomics Shared Resource (GSR) for 3′ end library preparation and sequencing. Using the Lexogen QuantSeq 3′ REV kit, 75 bp single-end (SE) strand-specific libraries were prepared, multiplexed, and sequenced on an Illumina Mini-Seq. For each sample, 6.92 ± 0.30 million reads were obtained, and read quality was confirmed using FastQC. Raw FASTQ files were aligned to the *Neurospora crassa* OR74A NC12 genome (FungiDB version 45; accessed October 25, 2019) using STAR [127]. Around 91.5% to 93.5% of the reads mapped uniquely to the NC12 genome. Because 3′ end libraries generate only 1 sequencing read at the extreme 3′ end of a given mRNA transcript (directly before its poly(A) tail), gene expression was quantified by counting reads assigned to each genetic locus using HTSeq-count [128]. Gene count normalization by library size between samples was performed using a custom R script. 3′ End Sequencing data have been submitted to the NCBI Gene Expression Omnibus (GEO; https://www.ncbi.nlm.nih.gov/geo/) under accession number GSE201901.

RNA-Sequencing datasets from 4 other studies were mined in the analyses presented. To examine wild-type *Neurospora* gene expression under carbon starvation (S4 Fig and S2 Table), RNA-seq data were taken from a study where liquid cultures (25˚C, LL) were grown for 16 hours in 1X Vogel's 2% sucrose minimal medium and shifted to either 0% or 2% glucose 1X Vogel's medium for 60 minutes [55] (GSE78952). To examine gene expression in the Δ*set-1* mutant background (Fig 5B), RNA-seq data were taken from a study where liquid cultures (25˚C, DD24) were grown in 2% glucose Liquid Culture Medium for 48 hours of total growth [71] (GSE121356). To examine gene expression in the Δ*set-2* mutant (Fig 5C), RNA-seq data were taken from a study where liquid cultures (32˚C) were grown in 1X Vogel's 1.5% sucrose medium [73] (GSE82222 and GSE118495). These three RNA-Seq datasets were reprocessed exactly as previously described [52], and FPKM gene expression values were used in the analyses presented. To examine and compare wild-type poly(A) tail locations (S3 Table), 2P-Seq data were taken from a previous study where nuclear fractions were isolated from 1X Vogel's 2% glucose liquid medium [65] (SRA PRJNA419320). Raw 2P-Seq data were filtered according to custom a Perl script from the original study ("Step 1": https://github.com/elifesciences-publications/poly-A-seq). After read filtering, duplicate 2P-Seq FASTQ files were processed in exactly the same manner as the new 3′ End Sequencing dataset generated in this study.

NC12 mapped reads from 3′ End Sequencing (this study) and 2P-Seq [65] data were sorted and indexed using Samtools (BAM file outputs) and then visualized using IGV. Read pileups denoted the location of poly(A) tails in both datasets. To map locations of poly(A) tails genome wide, the ChIP-Seq peak calling algorithm MACS2 was repurposed [129]. The relevant MACS2 parameters used to identify poly(A) peaks were as follows: effective genome size (-g 4.014e7), retention of duplicate reads in pileups (—keep-dup all), summit and subpeak identification (—call-summits), fragment size estimation and shifting turned off (—nomodel— extsize 75), and a false discovery rate cutoff for significant peaks (-q 0.01). MACS2 peaks were

assigned to the corresponding gene 3′ UTR region using a custom R script. The *Neurospora crassa* NC12 transcriptome annotation remains partially incomplete with only 7,793 out of 10,591 unique NCU IDs having 3′ UTRs annotated. As a result, approximately 14% to 18% of all MACS2 peaks were pruned from consideration due to missing annotations. Importantly, there are also examples of under-annotated 3′ UTR regions, where the poly(A) read pileup signal is clearly located outside of the 3′ UTR annotation. One such critical example occurs at the *frequency* locus (positive/Watson strand gene), where the predominant poly(A) peak is centered at LG VII coordinate 3,136,633, and its longest 3′ UTR annotation ends at coordinate 3,136,464. The *Neurospora* NC12 transcriptome annotation was last updated in March 2015, before migration from the Broad Institute to the FungiDB database [47]. The 3′ End Sequencing analyses presented here can be updated upon release of an improved transcriptome annotation. Furthermore, there are 621 instances of overlapping coordinates within the 3′ UTRs of tail-to-tail oriented genes, and any poly(A) peaks falling in these gene assignment ambiguous regions were also removed from consideration (approximately 9% of MACS2 peaks). The remaining MACS2 peaks (approximately 6,600 unique poly(A) peaks per sample) were assigned to the corresponding gene 3′ UTR region and analyzed using custom R scripts. APA events were defined as instances of more than one distinct MACS2 peak assigned to a single 3′ UTR region. poly(A) tail read pileups from genes of interest were extracted using the igvtools count function, and genome coordinate plots were generated using the R Bioconductor package Gviz (Fig 4C). 3′ UTR heatmaps were generated using a custom R script (Fig 4B). All custom R scripts for gene expression analyses and APA analyses are available at: https://github.com/cmk35.

## Protein isolation and detection

Frozen *Neurospora* tissue was ground in liquid nitrogen with a mortar and pestle. Total protein was extracted in buffer (50 mM HEPES (pH 7.4), 137 mM NaCl, 10% glycerol v/v, 0.4% NP-40 v/v, and cOmplete Protease Inhibitor Tablet according to instructions for Roche # 11 836 170 001) and processed as described [130]. Protein concentrations were determined by Bradford Assay (Bio-Rad # 500–0006). For western blots, equal amounts of total protein (30 μg) were loaded per lane into 4% to 12% Bis-Tris Bolt gels (Invitrogen # NW04125/NW04127) or 8% Bis-Tris Bolt gels (Invitrogen # NW00085/NW00087). Western transfer was performed using the Mini Blot Module (Invitrogen # B1000) and BOLT Transfer Buffer (Invitrogen # BT0006) onto an Immobilon-P PVDF membrane (Millipore # IPVH00010). Primary antibodies used for western blotting were anti-CK1a (1:2,000; rabbit raised) [52], anti-Tubulin alpha (1:5,000; Sigma # T6199), anti-WC-2 (1:5,000; rabbit raised) [131], and anti-FRH (1:10,000; rabbit raised) [77]. The secondary antibodies, goat anti-mouse or goat anti-rabbit HRP, were used at 1:5,000 (Bio-Rad # 170–6516, # 170–6515). SuperSignal West Pico PLUS Chemiluminescent Substrate (ThermoFisher # 34578) or Femto Maximum Sensitivity Substrate (Thermo # 34095) was used for detection. Immunoblot quantification and normalization were performed in ImageJ.

## Mammalian cell culture, synchronization, and siRNA knockdown reagents

U2OS cells were stably transfected under puromycin selection using a construct containing the mouse *BMAL1* promoter [132,133] driving destabilized luciferase [134]. U2OS-m*BMAL1-dLuc*-Puro (referred to as: *Bmal1-dLuc*) cells were maintained at 37°C and 6% $CO_2$ in 25 mM (high) glucose DMEM (Thermo Fisher # 11995–065 with 1 mM sodium pyruvate; or Thermo Fisher # 11965–092 without pyruvate) supplemented with 10% v/v FBS (Thermo Fisher # 10437–036, LOT # 2199672RP) and 1.5 μg/ml of puromycin (Sigma # P9620, 10 mg/ml stock).

For control Nutritional Compensation assays (S8 Fig), *Bmal1-dLuc* cells were subcultured from the same 100-mm dish and grown to 95% to 100% confluence in 35-mm dishes (Corning # 430165) containing 2 ml of DMEM 25 mM glucose, 10% FBS, and puromycin. Confluent cells were washed once in warm 1X PBS (pH 7.4) (Corning # 21-040-CV). The medium was changed to either 2 ml of DMEM 25 mM high glucose (Thermo Fisher # 11995–065 with 1 mM sodium pyruvate) or 2 ml of DMEM 5.56 mM low glucose (Thermo Fisher # 11885–084 with 1 mM sodium pyruvate). Both synchronization-release medium formulations were pre-warmed and each contained 10% v/v FBS, 1.5 µg/ml puromycin, 0.1 mM luciferin (GoldBio, 0.1 M stock), and 0.1 µM dexamethasone (Sigma # D2915, 1 mM stock). Dexamethasone is used to reset cells to the same circadian phase and initiate the circadian free run for recording.

For siRNA knockdown assays (Fig 6), *Bmal1-dLuc* cells were subcultured from the same 100-mm dish and grown to 60% to 80% confluence in 35-mm dishes containing DMEM 25 mM glucose, 10% FBS, and puromycin. Cells were washed once in 1X PBS, and the medium was changed to 2 ml of Opti-MEM (Thermo Fisher # 31985–070) with 5% v/v FBS. Cells were transfected with the indicated siRNAs (15 pmol of total siRNA per 35-mm dish) [85] using the Lipofectamine 3000 transfection reagent (Thermo Fisher # L3000) and according to the manufacturer's instructions for 6-well plates. Although the Opti-MEM medium formulation is not publicly available, one study reported the Opti-MEM glucose concentration as 2.5 g/L or 13.88 mM [135]. siRNAs were obtained from Qiagen: AllStars Negative Control siRNA (Qiagen # 1027280); human *SETD2* siRNA (Qiagen # 1027416, FlexiTube GeneSolution GS29072, 4× siRNAs used at 3.75 pmol each); human *CPSF6* siRNA (Qiagen # 1027416, FlexiTube GeneSolution GS11052, 4× siRNAs used at 3.75 pmol each); human *CRY2* siRNA (Qiagen # 1027416, FlexiTube GeneSolution GS1408, 4× siRNAs used at 3.75 pmol each) [86]. Cells were incubated for 2 days before removing the siRNA transfection medium and proceeding with RNA extraction or with circadian recordings.

RT-qPCR was used to validate siRNA knockdown efficiencies. Two-day transfected cells were washed once in 1 ml of ice-cold 1X PBS. Cells were harvested by scraping in 1 ml of TRIzol (Invitrogen), and total RNA extraction was performed according to the manufacturer's instructions. Approximately 500 ng of mRNA was converted into cDNA using the oligo(dT) method from the SuperScript IV First-Strand synthesis kit (Invitrogen # 18091–050). RT-qPCR was performed using SYBR green master mix (Qiagen # 204054) and a StepOne Plus Real-Time PCR System (Applied Biosystems). $C_t$ values were determined using StepOne software (Life Technologies) and normalized to the *GAPDH* gene ($\Delta C_t$). The $\Delta\Delta C_t$ method was used to determine mRNA levels relative to nontransfected negative control samples. Relevant RT-qPCR primer sequences are as follows: h*GAPDH*: 5′ TGCACCACCAACTGCTTAGC 3′ and 5′ ACAGTCTTCTGGGTGGCAGTG 3′. h*CPSF6*: 5′ GATGTGGGTAAAGGAGCAG 3′ and 5′ CTTCATCTGTTGTCCACCA 3′. h*SETD2*: 5′ CTTTCTGTCCCACCCCTGTC 3′ and 5′ CCTTGCACCTCTGATGGCTT 3′.

Two-day transfected cells were washed in warm 1X PBS and prepared for circadian synchronization. Synchronization-release medium was prewarmed and contained 1% v/v FBS, 1.5 µg/ml puromycin, 0.1 mM luciferin, and 0.1 µM dexamethasone. siRNA assays in DMEM were conducted using 2 ml of DMEM (Thermo Fisher # 11966–025) supplemented with 10 mM (low glucose) or 30 mM (high glucose) from a D-glucose stock solution (Sigma # G8644, 100 g/L stock). DMEM base medium contains more total nutrients than MEM—approximately 2-fold higher levels of the 13 essential amino acids, about 4-fold higher levels of the 8 vitamins, and includes the nonessential amino acids (Gly and Ser) in its formulation. siRNA assays in MEM were conducted using 2 ml of MEM (Thermo Fisher # 11095–080) containing 5.56 mM (low glucose) or supplemented up to 25 mM (high glucose) from a D-glucose stock solution (Sigma). Unlike *Neurospora*, complete glucose starvation medium did not support

cell viability in preliminary experiments using DMEM medium containing 10% v/v FBS but zero additional glucose. "Low" 5 to 10 mM glucose was defined by manufacturer formulations as well as physiological levels of fasting serum glucose in humans.

## Mammalian luciferase reporter detection and period length calculations

Immediately prior to bioluminescent recording, *Bmal1-dLuc* cells in 35-mm dishes were covered with 40-mm circular microscope cover glass (Fisher Scientific # 22038999 40CIR-1) and sealed using high-vacuum silicone grease (Dow Corning # Z273554). Luciferase data were collected in a LumiCycle 32 (ActiMetrics) luminometer every 10 minutes for 5 to 6 days. Raw luciferase traces in bioluminescence counts/second units were exported using LumiCycle analysis software (ActiMetrics, version 2.56). Data from individual plates were manually combined and converted to hours postsynchronization using Microsoft Excel. Period lengths for each luciferase trace were calculated using 3 different methods and averaging the period results with equal weights. For the first method, signal peaks and troughs were extracted from days 1 to 3 of raw data, and period was estimated by subtracting consecutive peaks or troughs as described [136]. Second, the WaveClock algorithm was implemented in R [137]. Finally, the suite of *Neurospora* period length tools was used as previously described [52]. For *Bmal1-dLuc* luciferase trace data visualization purposes (Fig 6 and S8 Fig), raw counts per second values sampled within the same hour were averaged together (i.e., data were down-sampled from 10-minute measurement intervals to 1-hour measurement intervals).

## Data visualization

All figures were plotted in R, output as scalable vector graphics, formatted using Inkscape, and archived in R markdown format. Data represent the mean of at least three biological replicates with ±1 standard deviation error bars, unless otherwise indicated. All statistical tests were performed in R.

## Supporting information

**S1 Fig. Additional properties of Nutritional Compensation in *Neurospora crassa*.** A fungal biomass control was implemented to ask whether the amplitude of the core clock transcriptional reporter changes with glucose levels (e.g., Fig 1B). The *gpd* promoter from *Cochliobolus heterostrophus* driving constitutive *luciferase* was used as a reporter for fungal biomass. Not surprisingly, biomass increases as a function of glucose (N = 3 race tubes per glucose concentration). Many traces showed a decrease in bioluminescence at approximately 100 hours, and this correlates with the fungal growth front reaching and surpassing the end of the device's recording area (**A**) (see S1 Movie). Averaged replicates are shown for the *frq* clock box transcriptional reporter across all glucose levels (N = 6; expanded Fig 1B) (**B**). Detrended clock reporter traces were plotted on a circadian time (CT) scale to normalize for the slight period differences (Fig 1A). Circadian phase is consistent across glucose levels (**C**). Amplitude was computed for each individual biomass reporter trace using data from hours 25–108 (amplitude calculation: [maximum value − minimum value] / 2) (**D**). Average amplitude was computed for each individual core clock reporter trace using data from days 2–5 (hours 25–112) to extract 4 peak and 4 trough values. The first day (hours 0–24) was omitted due to low fungal biomass and consequently low luciferase signal during the first recording day (**E**). Core clock amplitudes were normalized to biomass by computing the amplitude ratio at the respective glucose concentrations. There is no clear increasing or decreasing trend of normalized core clock amplitude as a function of glucose, and, therefore, the higher magnitude oscillations observed at high glucose concentrations are most likely a function of increased biomass only

(**F**). Growth rates were computed from biomass reporter experiments by estimating the linear growth rate at 5 consecutive 12-hour intervals from 36–84 hours in constant darkness (**G**). No arginine was added to the race tube medium for any experiment shown. The raw data underlying parts (**A**–**G**) can be found in S7 Data.
(EPS)

**S2 Fig. A subset of FGSC knockout strains were identified with strong conidial banding phenotypes on glucose starvation medium.** A representative race tube from the primary genetic screen is shown. Six out of 7 knockout strains with strong banding phenotypes have a wild-type circadian period length and normal compensation, except for ΔNCU06565 (FGSC15333; S3 Fig). Like the *ras-1*$^{bd}$ (NCU08823) point mutant, all knockout strains have a reduced linear growth rate compared to the wild-type control FGSC2489. FGSC = Fungal Genetics Stock Center, curator of the *Neurospora* whole genome knockout collection.
(EPS)

**S3 Fig. Six out of 12 new nutritional under-compensation mutants, which lack clear GO Term relationships.** Circadian bioluminescence was recorded from race tube cultures of the indicated deletion mutants. High nutrient medium (yellow lines) contained 0.5% w/v glucose 0.17% w/v arginine, and zero nutrient medium (blue lines) contained 0% glucose 0% arginine (standard deviation error bars). Period lengths were computed ($N \geq 2$ biological replicate period estimates per nutrient concentration) and summarized in a bar graph compared to controls. "Het" indicates heterokaryon strains derived from the *Neurospora* whole genome deletion collection, which were maintained on hygromycin selection medium prior to the bioluminescence race tube assays. FGSC = Fungal Genetics Stock Center, curator of the *Neurospora* whole genome knockout collection. The raw data underlying this Figure can be found in S8 Data.
(EPS)

**S4 Fig. Core clock and compensation gene expression upon glucose starvation.** RNA-Sequencing data were mined from a previous study [55] (Materials and methods), where liquid cultures were either maintained in 2% glucose or shifted to glucose starvation for 60 minutes. Duplicate transcriptomes from the 2% glucose condition were compared to 0% glucose starvation replicates, Z-scores were computed for the 8,796 expressed genes in the dataset, and *prd-1* (NCU07839) and *frh* (NCU03363) were found among the top 220 genes (top 2.5%) in the entire transcriptome down-regulated after glucose starvation. The raw data underlying this Figure can be found in S9 Data.
(EPS)

**S5 Fig. Solid medium cellophane plate cultures maintain circadian function and recapitulate Nutritional Compensation phenotypes of interest.** Liquid cultures containing wild-type fungal plugs (1.8% glucose) and cellophane plate cultures inoculated with wild-type conidia (0.5% w/v glucose, 0.17% w/v arginine) were set up concurrently. Liquid and solid cultures were entrained in constant light at 25°C for at least 16 hours, and serial light-to-dark transfers were performed to sample 1.5 cycles of circadian time points from DD4 to DD28 (4-hour sampling density, 44–48 hour total culture ages). Total RNA was isolated from each time course sample, and *frq* mRNA rhythms were examined by northern blot ($N = 1$ time course replicate). RNA levels were quantified using ImageJ densitometry, normalized, and plotted as line graphs. The circadian clock is clearly functional in both growth regimes (**A**). Circadian bioluminescence was recorded from cellophane plate cultures of the indicated genotypes grown on high nutrient medium (0.25% w/v glucose, 0.17% w/v arginine). One representative luciferase trace is shown from $N = 2$ biological replicates per strain. Period lengths were calculated, and results

agree with Nutritional Compensation phenotypes derived from the race tube screen (S1 Table): control: 20.6 ± 0.3 hours; Δcpsf5 Δcpsf6 double mutant: 17.5 ± 0.4 hours (**B**). Circadian bioluminescence was recorded from cellophane plate cultures of the indicated genotypes grown on zero nutrient medium (0% glucose, 0% arginine). One representative luciferase trace is shown from $N$ = 2 biological replicates per strain. Period lengths were calculated, and results agree with Nutritional Compensation phenotypes derived from the race tube screen (S1 Table): control: 22.0 ± 0.6 hours; Δset-2: 20.5 ± 0.1 hours (**C**). The raw data underlying parts (**A**–**C**) can be found in S10 Data.
(EPS)

**S6 Fig. Representative changes in the APA landscape (50% of genes) in Δcpsf5 Δcpsf6 double mutants compared to controls.** Integrative Genomics Viewer (IGV) screenshots are shown for 4 example 3′ UTR regions. NCU01154 (*sub-1*) is 1 out of the 193 APA events in controls collapsing to a single poly(A) peak in mutants (**A**). NCU02435 (histone H2B) is 1 out of the 123 APA events in mutants collapsing to a single poly(A) peak in controls (**B**). NCU01418 (*ccg-6*) and NCU16757 are 2 examples out of the 155 APA events in control and in mutant where the location of the predominant poly(A) peak was significantly changed (**C**). The majority of the 155 predominant poly(A) peak changes involve a distal-to-proximal peak shift in mutant (e.g., NCU01418) and a minority involve a proximal-to-distal peak shift (e.g., NCU16757).
(EPS)

**S7 Fig. Removal of COMPASS complex subunit *swd3* phenocopies the long period length and Nutritional Compensation defects of Δset-1, indicating that SET-1 and the COMPASS complex are required for normal compensation.** The 96-well plate luciferase assays were used to measure the circadian period length of the indicated COMPASS complex subunit knockout mutants in constant darkness at 25°C on standard medium. Averaged replicates are shown for Δsdc1 ($N$ = 3 biological replicates), Δswd1 ($N$ = 2 biological replicates), and Δswd2 ($N$ = 3 biological replicates) (standard deviation error bars). The table (right) summarizes circadian phenotypes of COMPASS complex subunit knockouts in this study compared to previous work [70] (**A**). Averaged replicates are shown for 96-well plate luciferase assays of Δset-1 ($N$ = 3 biological replicates), Δswd3 ($N$ = 4 biological replicates), and Δset-1 Δswd3 ($N$ = 2 biological replicates) (standard deviation error bars). Period lengths were computed and summarized in a bar graph. All 96-well plate biological replicates contained $N$ = 3 technical replicates per genotype (**B**). Circadian bioluminescence was recorded from race tube cultures of the indicated knockout mutants. High nutrient medium (yellow lines) contained 0.5% w/v glucose 0.17% w/v arginine, and zero nutrient medium (blue lines) contained 0% glucose 0% arginine (standard deviation error bars). Period lengths were computed for Δset-1 ($N$ = 4 biological replicates per nutrient condition), Δswd3 ($N$ = 2 biological replicates), and Δset-1 Δswd3 ($N$ = 2 biological replicates) and summarized in a bar graph compared to the control (standard deviation error bars) (**C**). The raw data underlying parts (**A**–**C**) can be found in S11 Data.
(EPS)

**S8 Fig. *Bmal1-dLuc* cells are slightly under-compensated from 5.56 mM low glucose to 25 mM high glucose.** U2OS cells were synchronized using dexamethasone and released into DMEM low or high glucose medium. Representative luciferase traces are shown from one biological replicate with $N$ = 4 technical replicates per glucose concentration (standard deviation error bars) (**A**). Period lengths were calculated for 25–26 total replicates per glucose concentration and plotted as a boxplot. As observed in *Neurospora*, the human circadian period length is under-compensated and shortens slightly with increasing glucose ($p < 0.01$, Student $t$ test).

Average period lengths were 24.5 ± 0.7 hours (low glucose) and 24.0 ± 0.4 hours (high glucose) (**B**). The raw data underlying parts (**A**) and (**B**) can be found in S12 Data.
(EPS)

**S1 Table. Genetic screen for Nutritional Compensation defects.** Results are presented from the 3-phase genetic screen for Nutritional Compensation defects among *Neurospora* knockout strains. Period lengths are shown for each knockout strain. Knockout strains were retained through each phase of the screen if circadian period changes were observed relative to the wild-type control or if a clear lack of circadian banding pattern was observed in the primary screen ("output" defective mutants). Nutritional under-compensation mutants were defined by ratios of high-to-zero glucose period lengths ≤ 0.90 (pink font). Nutritional over-compensation mutants had period length ratios ≥ 1.02 (green font).
(XLSX)

**S2 Table. Informatic description of *Neurospora* genes screened for Nutritional Compensation defects.** The first tab of the Table is filtered for gene knockouts that progressed to the tertiary genetic screen, and the second tab of the Table contains all novel genes screened for compensation defects. Circadian rhythmicity at the gene and protein level among knockouts screened was determined from previous studies [45,138]. Promoter binding by the WCC positive arm transcription factors was determined from previous studies [138–140]. Light-regulated gene activation or repression was determined from previous studies [139,141,142]. Gene expression Z-scores from carbon starvation conditions were also reported [55] (S4 Fig).
(XLSX)

**S3 Table. Consensus list of *Neurospora* genes with Alternative Polyadenylation (APA) in 3′ UTRs.** A total of 843 genes contain multiple poly(A) sites within 3′ UTR regions from the intersection of this study and previous work [65]. Nine out of 843 genes with 3′ UTR APA are highlighted as core clock genes or compensation screen hits. Annotated MACS2 poly(A) peak results are shown for the 843 genes with APA from each dataset as individual tabs.
(XLSX)

**S4 Table. Alternative Polyadenylation (APA) events altered in the ΔCFIm knockout mutant compared to controls.** A total of 1,447 total genes contain multiple poly(A) sites within 3′ UTR regions in wild-type control and/or ΔCFIm mutant data from this study. In all 4 datasets, 940/1,447 genes display APA. Out of 1,447 instances, 123 are recorded where a single poly(A) peak in control expands to multiple APA events in mutant (dark green highlight). Out of 1,447 instances, 193 are reported of a single poly(A) peak in mutant expanding to multiple APA events in wild-type controls (light green highlight). Out of 1,447 APA events, 155 occur where the location of the predominant poly(A) peak was significantly changed in the mutant background (orange highlight). Annotated MACS2 poly(A) peak results are shown for the APA genes from each dataset as individual tabs.
(XLSX)

**S1 Movie. Nutritional Compensation properties of wild-type *Neurospora crassa*.** Circadian bioluminescence of exemplar wild-type race tubes (Fig 1) is shown for zero nutrient medium (blue line; 0% glucose 0% arginine) (**A**) compared to high nutrient medium (yellow line; 0.5% w/v glucose 0% arginine) (**B**). Average period lengths are 21.9 ± 0.5 hours for 0% glucose and 20.8 ± 0.2 hours for 0.5% w/v glucose (average ± 1 SD).
(MOV)

**S2 Movie. Nutritional Compensation defect in the Δ*cpsf6* mutant.** Circadian bioluminescence of one exemplar Δ*cpsf6* race tube (Fig 4A) is shown for whole-tube quantification of

high amino acid medium growth (yellow line; 0% glucose 0.17% w/v arginine; average period = 19.4 ± 0.3 hours) (**A**) compared to old tissue quantification (blue line; 0% glucose 0.17% w/v arginine; period = 17.3 ± 0.3 hours) (**B**). This approximately 2-hour period difference indicates that Δ*cpsf* mutants must undergo a transition from high-to-low amino acid levels to reveal the Nutritional Compensation defect.
(MOV)

**S3 Movie. Nutritional Compensation defect in the Δ*set-2* mutant.** Circadian bioluminescence of one exemplar Δ*set-2* race tube (Fig 5A) out of 4 biological replicates. Rhythms on zero nutrient medium (blue line; 0% glucose 0% arginine) (**A**) are compared to high nutrient medium (yellow line; 0.5% w/v glucose 0% w/v arginine) (**B**). Average period lengths are 20.3 ± 0.7 hours for 0% glucose and arrhythmic for high nutrients.
(MOV)

**S1 Data. Raw data accompanying main text Fig 1.** The first tab of the Data File corresponds to period length measurements for Fig 1A. The second and third tabs of the Data File correspond to raw and processed traces for Fig 1B.
(XLSX)

**S2 Data. Raw data accompanying main text Fig 2.** The first tab of the Data File corresponds to temperature compensation period length measurements for Fig 2A. The second and third tabs of the Data File correspond to raw and processed traces for Fig 2B.
(XLSX)

**S3 Data. Raw data accompanying main text Fig 3.** The first and second tabs of the Data File correspond to raw and processed traces for Fig 3A. The third, fourth, and fifth tabs of the Data File correspond to raw and processed traces for Fig 3C as well as circadian period lengths. The sixth tab of the Data File corresponds to immunoblot densitometry measurements derived from ImageJ for Fig 3D.
(XLSX)

**S4 Data. Raw data accompanying main text Fig 4.** The first, second, and third tabs of the Data File correspond to raw and processed traces for Fig 4A as well as circadian period lengths. The fourth tab of the Data File corresponds to averaged and $\log_2$-normalized read counts of core clock and compensation relevant genes for Fig 4D.
(XLSX)

**S5 Data. Raw data accompanying main text Fig 5.** The first and second tabs of the Data File correspond to raw and processed traces for Fig 5A. The third tab of the Data File corresponds to averaged and $\log_2$-normalized fpkm values of core clock genes for Fig 5B. The fourth tab of the Data File corresponds to averaged and $\log_2$-normalized fpkm values of core clock genes for Fig 5C. The fifth tab of the Data File corresponds to RT-qPCR values for Fig 5D. The sixth tab of the Data File corresponds to race tube period lengths for Fig 5E.
(XLSX)

**S6 Data. Raw data accompanying main text Fig 6.** The first, second, and third tabs of the Data File correspond to raw and processed traces for Fig 6A as well as circadian period lengths. The fourth tab of the Data File corresponds to RT-qPCR values for Fig 6A. The fifth, sixth, and seventh tabs of the Data File correspond to raw and processed traces for Fig 6B as well as circadian period lengths.
(XLSX)

**S7 Data. Raw data accompanying S1 Fig.** The first and second tabs of the Data File correspond to raw and processed traces for S1A Fig. The third and fourth tabs of the Data File correspond to raw and processed traces for S1B Fig. The fifth tab of the Data File corresponds to detrended traces on a circadian time (CT) scale for S1C Fig. The sixth tab of the Data File corresponds to average amplitude measurements and ratios for S1D–S1F Fig. The seventh tab of the Data File corresponds to growth rate measurements for S1G Fig.
(XLSX)

**S8 Data. Raw data accompanying S3 Fig.** The first and second tabs of the Data File correspond to raw and processed traces for S3 Fig.
(XLSX)

**S9 Data. Raw data accompanying S4 Fig.** The first tab of the Data File corresponds to normalized fpkm values of core clock and compensation relevant genes with Z-score comparing starvation to glucose replete conditions for S4 Fig.
(XLSX)

**S10 Data. Raw data accompanying S5 Fig.** The first tab of the Data File corresponds to immunoblot densitometry measurements derived from ImageJ for S5A Fig. The second tab of the Data File corresponds to raw traces and circadian period lengths for S5B Fig. The third tab of the Data File corresponds to raw traces and circadian period lengths for S5C Fig.
(XLSX)

**S11 Data. Raw data accompanying S7 Fig.** The first, second, and third tabs of the Data File correspond to raw and processed traces as well as circadian period lengths for S7A Fig. The fourth, fifth, and sixth tabs of the Data File correspond to raw and processed traces as well as circadian period lengths for S7B Fig. The seventh and eighth tabs of the Data File correspond to raw and processed traces as well as circadian period lengths for S7C Fig.
(XLSX)

**S12 Data. Raw data accompanying S8 Fig.** The first and second tabs of the Data File correspond to raw and processed traces for S8A Fig. The third tab of the Data File corresponds to circadian period lengths for S8B Fig.
(XLSX)

**S1 Raw Images. Raw immunoblot data accompanying Fig 3D and S5 Fig.** All immunoblots are included with 1 or 2 TIF images each showing raw blots uncropped and labeled with antibody used, method, sample ID, and molecular weight markers (when applicable). "X"s mark lanes and samples that were not shown in Fig 3D.
(PDF)

## Acknowledgments

We thank the Fungal Genetics Stock Center (Kansas City, Missouri, USA) for curating *N. crassa* strains and the whole genome deletion collection (P01 GM068087). We thank Adrienne Mehalow and Arko Dasgupta for generously sharing the collection of *Neurospora* kinase knockout circadian reporter strains, Bradley Bartholomai for the constitutive promoter driving luciferase construct, Consuelo Olivares-Yañez and Alejandra Goity from the Luis Larrondo laboratory at the Pontificia Universidad Católica de Chile for their virtual assistance with the cellophane plate growth method, Lauren Francey from the John Hogenesch laboratory at Cincinnati Children's Hospital Medical Center for virtual assistance with the U2OS siRNA knockdown experiments, Chris Baker at the Jackson Laboratory for discussions on the *Neurospora*

CFIm complex, Josh Gamsby at the University of South Florida for the *Bmal1-dLuc* stable cell line, and Christine Mayr at Memorial Sloan Kettering Cancer Center for discussions and mentorship on the Alternative Polyadenylation (APA) work. 3′ End Sequencing was carried out at the Geisel School of Medicine at Dartmouth in the Genomics Shared Resource (GSR) in collaboration with Fred Kolling IV. Dartmouth GSR is supported by NIH and NSF equipment grants as well as an NCI Cancer Center Core Grant (P30 CA023108) and an NIGMS COBRE Grant (P20 GM130454).

## Author Contributions

**Conceptualization:** Christina M. Kelliher, Elizabeth-Lauren Stevenson, Jennifer J. Loros, Jay C. Dunlap.

**Data curation:** Christina M. Kelliher, Elizabeth-Lauren Stevenson.

**Formal analysis:** Christina M. Kelliher, Elizabeth-Lauren Stevenson.

**Funding acquisition:** Christina M. Kelliher, Jennifer J. Loros, Jay C. Dunlap.

**Investigation:** Christina M. Kelliher, Elizabeth-Lauren Stevenson.

**Methodology:** Christina M. Kelliher.

**Resources:** Jennifer J. Loros, Jay C. Dunlap.

**Software:** Christina M. Kelliher.

**Supervision:** Jennifer J. Loros, Jay C. Dunlap.

**Validation:** Christina M. Kelliher.

**Visualization:** Christina M. Kelliher, Elizabeth-Lauren Stevenson.

**Writing – original draft:** Christina M. Kelliher.

**Writing – review & editing:** Christina M. Kelliher, Jennifer J. Loros, Jay C. Dunlap.

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
