## [Editor Report · Decision Letter 0]

23 May 2022

Dear Dr Kelliher, 

Thank you for submitting your manuscript entitled "A role for gene expression and mRNA stability in nutritional compensation of the circadian clock" for consideration as a Research Article by PLOS Biology, and apologies again for our delay in sending you an initial decision.

Having now had a chance to discuss your manuscript with the PLOS Biology editorial staff as well as with an academic editor with relevant expertise, and I am writing to let you know that we would like to send your submission out for external peer review.

Once your full submission is complete, your paper will undergo a series of checks in preparation for peer review. After your manuscript has passed the checks it will be sent out for review. To provide the metadata for your submission, please Login to Editorial Manager (https://www.editorialmanager.com/pbiology) within two working days, i.e. by May 25 2022 11:59PM.

Kind regards,

Lucas

Lucas Smith

Associate Editor

PLOS Biology

lsmith@plos.org

---

## [Decision Letter · Decision Letter 1]

23 Jun 2022

Dear Tina,

Thank you again for your patience while your manuscript "A role for gene expression and mRNA stability in nutritional compensation of the circadian clock" was peer-reviewed at PLOS Biology. It has now been evaluated by the PLOS Biology editors, an Academic Editor with relevant expertise, and by several independent reviewers. 

In light of the reviews, which you will find at the end of this email, we would like to invite you to revise the work to thoroughly address the reviewers' reports. 

As you will see from their comments, all of the reviewers appreciated the importance of the topic examined here, and Reviewer 1 has suggested that we accept the manuscript for publication. However, Reviewers 2 and 3 raise a number of important concerns which would need to be carefully and thoroughly addressed, with new data where needed, before we can consider your manuscript for publication. In particular, the reviewers have highlighted conclusions that are not yet adequately supported by the data, noting that additional work is needed to support the role of the identified candidate genes in nutritional compensation and your conclusion that this is a conserved process.

Given the extent of revision needed, we cannot make a decision about publication until we have seen the revised manuscript and your response to the reviewers' comments. Your revised manuscript is likely to be sent for further evaluation by all or a subset of the reviewers.

**IMPORTANT - SUBMITTING YOUR REVISION**

*Re-submission Checklist*

*Published Peer Review*

*PLOS Data Policy*

*Blot and Gel Data Policy*

Sincerely,

Luke

Lucas Smith, Ph.D.

Associate Editor

PLOS Biology

lsmith@plos.org

REVIEWS:

Reviewer #1: This is a very solid and comprehensive study of the roles of new regulators of Nutritional Compensation in filamentous fungus Neurospora crassa and in human cells. By performing a genetic screen under glucose and amino acid starvation conditions, the authors showed that 16 new mutants play different roles in regulating gene expression and mRNA stability in nutritional compensation of circadian clock in Neurospora crassa. Surprisingly, they found that knockdown of CPSF6 or SETD2 in human cells also results in nutrient-dependent period length changes. Together, these results prove that Nutritional Compensation is a conserve circadian process in fungal and mammalian clock and suggest that it may share similar molecular mechanism. Despite the large amount of work, the quality of the results is high and the conclusions are well support. I think it is suitable for publication in PLOS BIOLOGY though I have a minor question.

Are these 16 factors also involved in the nutritional compensation besides glucose?

Reviewer #2: A fundamental property of circadian clocks is their ability to keep time over abroad range of physiological conditions. The underlying mechanisms are purely understood. Kelliher et al performed an elaborate three-step screen in Neurospora to identify genes involved in nutritional compensation of the circadian clock. They then further analyze two candidate genes, set2 and cpsf6, respectively. Downregulation of the mammalian homologs of these genes impacts NC of the clock of cultured mammalian cells, implying that mechanisms underlying NC of the circadian clock are similar in fungi and mammals. I find that the work needs more data supporting the role of the identified candidate genes in NC. 

Particularly, the Neurospora ∆set2 strain looks rhythmic to me (Fig. 5A) and period length is potentially not altered between low and high glc/arg levels. This has to be thoroughly examined. Neither did ∆cpsf6, by the author's definition, have a large enough phenotype to be considered under-compensated. If the NC phenotype of ∆cpsf6 is real but subtle, more N's are needed to show that the difference is statistically significant. Furthermore, period length of ∆cpsf6, which is short in general, is affected by arg concentration but not by glc concentration, while knockdown of its mammalian homolog in cell culture induces a long period phenotype, which is glc sensitive. It begs the question if the role in nutritional compensation of these to genes and the underlying mechanisms are indeed similar in fungi and mammals.

Substantial work went into the data collection for this manuscript and the data gives direction as to where mechanisms for NC could be found. It is a bit disappointing that rather than conducting follow up experiments that would help to pinpoint how the identified genes affect the clock machinery, such experiments are mostly announced to be done in "future work". Data from published transcriptomes, as well as data from other publications, provide evidence that is partly circumstantial and partly contradictory. Potential connections are discussed but experiments showing or confirming a direct impact of the identified genes on NC of the clock are missing.

There are a number of issues that should be addressed in a revised version which I list by order of appearance (and not priority) in the manuscript:

Lines 103-105: There is a recent paper by Marzoll et al. (2022) showing that Neurospora FRQ as well as mammalian PER2 are phosphorylated by their respective CK1 in vitro on a circadian scale in a temperature compensated fashion. This should be cited here.

120-122: While general transcription repressor RCO-1 has been shown to downregulate frq transcription as cited (Zhou et al. 2013), importantly, together with RCM-1 it is also part of the CSP-1 repressor complex (Sancar et al 2011). ∆rco-1 has a more severe effect on rhythmic conidiation as well as period and amplitude of the clock than ∆csp-1, yet the metabolic compensation effect is not very different if one looks at the data presented in Olivares-Yanez et al, 2016. This should be considered and discussed also in the later sections where RCO-1 is mentioned (e.g. in the pkac-1 mutant context).

From Figure 2 ff: what are the error bars of the bioluminescence traces if N=2 in almost all cases? It might be better showing both traces and the average.

Figure 2C: Please expand the abbreviation "FGSC" in the figure legend or in the text describing the figure.

It would also improve clarity if the authors additionally mention here what kind of experiments were performed in the individual screens, e.g. 1st screen: conidiation rhythms on glucose starvation racetubes, 2nd screen: luciferase reporter assays on 96-well plates -Glc -Arg…

263ff and Supplementary Table 1. Here it gets a little bit confusing with the number of the under-compensated mutants if you look at table S1. There, 14 genes are marked as under-compensated when mutated. In line 263 the authors examine "the group of 12 under-compensated mutants". I know the other two are looked at more closely from 311ff but this is still confusing. Please rewrite for clarity.

Please mention that ∆upf1 and ∆upf2 are the two most significant hits in 265. Otherwise they would not be mentioned at all in the main text. 

Supplementary Table 1:

mek-1 is listed in the table as RNA binding and not as a (MAPK-)kinase.

Please write "Arrhythmic" instead of ARR.

Define "output" in primary screen.

Supplementary Table 2: The table is used to show potential clock or light regulation of the screen results. Therefore, for easier reading, only the potential genes involved in temperature compensation should be shown or at most the 52 strains plus controls from the tertiary screen.

277-279: I cannot follow the reasoning why the other under-compensation mutants were not pursued further as they do meet the period ratio criteria.

Figure 3: The prd-4 (S493L) mutant does not really belong here and the authors should consider taking it out completely as, in contrast to the other strains tested, it is a gain of function mutant. The Dunlap/Loros labs published quite some time ago that the mutation enhances interaction of PRD-4 with FRQ leading to phosphorylation of FRQ by this checkpoint kinase thereby accelerating the clock (Pregueiro et al, 2006). The prd-4 KO strain did not show altered nutritional compensation (see Supplementary Table 1). With the space gained the bar graph could be enlarged a little bit. 

299: It is very interesting that the CK1a 3'UTR mutant has a similar NC phenotype as ∆upf-1prd-6. To state that the phenotype of the ck1a gene mutant is "not as severe" as the phenotype of ∆upf-1prd-6, though, would require statistical analysis of a sufficiently large set of independent experiments, as this would be a rather important point the authors make here.

301: Figure 5B doesn't show this!

 Are wc-2 and frh both up-regulated in the NMD mutants?

301-310: Here, western blots and qRT-PCR data should be shown, confirming data from transcriptome analyses cited that in the mutants wc-2 and frh transcript as well as the respective protein levels are altered.

312: "compared to the 11 remaining under-compensation mutants". 11 in this case because the authors probably did not include the csp-1 "positive control" strain?! Please be consistent with the numbers. Moreover, cspf6 is marked in the table but doesn't meet the authors' criteria for under-compensation (yet is described later by the authors as "one of the most significant compensation phenotypes" line 541. This turns out to be a major issue; see below).

365-372: For comparison to Figure 4B it would be good to show representative examples for the 3 groups of changes in the APA landscape in the supplementary section.

378-429, Figure 4:

In this screen only two (of the handpicked) genes had altered APA in the double mutant. Apparently, however, this did not change gene expression levels of these genes. On the other hand, genes with altered expression levels in the double mutant did not exhibit an altered APA landscape in the assay. So, there is no correlation and thus no indication that control of APA has a direct effect on NC. Is there an effect on protein translation levels maybe?

Additionally, altered gene expression levels should have been confirmed via qRT-PCR.

438-439: briefly mention how PKA regulates RCO-1 (inactivation of the repressor by phosphorylation). Also see above regarding RCO-1. Observation is also compatible with inhibition of CSP-1/RCO-1/RCM-1 complex.

446-448: Is it completely arrhythmic? In supp Table 1 you write: 31.7+-0 // ARR. Does that mean it was rhythmic in one experiment? If it's not clear the experiment should be repeated with sufficient N's and data should be detrended to potentially uncover a masked rhythm (see also below).

463-505, Figure 5: In Figure 5A, to my eye, ∆set-2 oscillates at high nutrition levels with a similar period length as at low nutrition, albeit at low amplitude and potentially masked by the steeply increasing average bioluminescence. As above this experiment should be repeated in order to get data from more independent experiments and data should also be detrended to reveal potential rhythms. This is important as the authors use the mammalian homolog for their cell culture experiments in the assumption that this is good candidate involved in NC.

471: "Δset-2 mutant results in … high frq expression levels … (Zhou et al., 2013; Sun et al., 2016)." This observation is not reproduced in the ∆set-2 transcriptome dataset from Bicocca et al 2016 shown in Figure 5C.

Curiously FRQ protein levels in ∆set-2 were not shown to be elevated in Sun et al., 2016. They were actually slightly lower. The observed decrease in frh found by Bicocca could account for this but these data would need to be confirmed via qRT-PCR or western blots with the strains used in this work. 

472-474: The authors should back their claim and show that WCC-independent frq transcription is indeed nutrient dependent using the appropriate strains.

541-551: now here is where the narrative does not fit: As mentioned above ∆cpsf6 did not meet the threshold criteria the authors defined for mutants to be under-compensated. And the phenotype is only seen in old mycelia and in response to changes in arginine levels not glucose. So, it is hardly one of the most significant compensation phenotypes. For ∆set-2 the data did not convince me. See above. 

The most convincing, interesting and obvious candidates to investigate NC in cell culture would have been upf-1 and upf-2 as well as set-1 or prd-1. Why not analyze homologs of these?

So, while the authors did find an NC phenotype in the siRNA knockdown of the homolog genes, the reasoning behind picking these genes leaves me puzzled.

553: In Neurospora, ∆cpsf6 exhibited the NC phenotype in response to amino acid starvation and not glucose. Yet, in the cell culture experiments the knockdown of its mammalian homolog has a NC phenotype in response to changes in glucose levels. This is not discussed by the authors. 

It is also strange that knockout / knockdown of cpsf6 both exhibit an under-compensated clock phenotype, yet has the opposite effect on period length in cell culture. 

Figure 6: Are the differences in period length statistically significant for CPSF1 and SETD2 in 6A?

What are the error bars in the Bioluminescence traces in 6A?

563-564: looks more like the luciferase expression levels are higher than the amplitude of oscillation. See/compare also Fig 6B.

781-782: format reference 

Reviewer #3: In this manuscript, Kelliher et al reported a genetic screen for genes involved in nutritional compensation of the circadian clock. They successfully identified a number of novel regulators of nutritional compensation, including genes with known functions in RNA stability, mRNA cleavage and polyadenylation, and histone modifying enzymes. Beyond the screen, they performed additional experiments to demonstrate that: 1) the Neurospora CFIm complex, similar to its mammalian homologues, is involved in mRNA alternative polyadenylation; 2) CPSF6 and SETD2 also regulate nutritional compensation in mammalian cells. 

Overall this study described an interesting screen that identified a number of novel regulators of nutritional compensation of the circadian clock. The results are of broad interest. However, characterization of these regulatory factors are lacking or very limited. For example, the data quality for CFIm-mediated regulation of mRNA alternative polyadenylation seemed poor and no data is provided to suggest that CFIm-mediated regulation of mRNA 3' UTRs is involved in nutritional compensation. Overall this study seems quite promising, but is a bit premature in its current form for publication in Plos Biology.

---

## [Decision Letter · Decision Letter 2]

28 Nov 2022

Dear Dr Kelliher,

Thank you for your patience while we considered your revised manuscript "A role for gene expression and mRNA stability in nutritional compensation of the circadian clock" for publication as a Research Article at PLOS Biology. This revised version of your manuscript has been evaluated by the PLOS Biology editors, the Academic Editor and the original reviewers.

The reviews are appended below. As you will see, the reviewers are largely satisfied with the revision, although Reviewer 2 has raised two minor concerns. Therefore, we are likely to accept this manuscript for publication, provided you satisfactorily address the remaining points raised by Reviewer 2. 

**IMPORTANT: Additionally, as you address Reviewer 2's comments in a revised manuscript, we also ask that you attend to the following editorial requests: 

1) TITLE: After some discussion within the team, we are wondering if the title should be edited slightly to more strongly present the main findings of the paper. If you agree (and if you feel it is supported), we suggest that you change the title to something like ""Nutritional compensation of the circadian clock is a conserved process controlled by gene expression regulation and mRNA stability"

2) BLOT AND GEL REPORTING REQUIREMENTS: Please note that we require the original, uncropped and minimally adjusted images supporting all blot and gel results reported in an article's figures or Supporting Information files. We will require these files before a manuscript can be accepted so please prepare and upload them now. Please carefully read our guidelines for how to prepare and upload this data: https://journals.plos.org/plosbiology/s/figures#loc-blot-and-gel-reporting-requirements

Please provide the original uncropped images accompanying Fig 3D; Fig S5A;

3)DATA POLICY: You may be aware of the PLOS Data Policy, which requires that all data be made available without restriction: http://journals.plos.org/plosbiology/s/data-availability. For more information, please also see this editorial: http://dx.doi.org/10.1371/journal.pbio.1001797

a. Supplementary files (e.g., excel). Please ensure that all data files are uploaded as 'Supporting Information' and are invariably referred to (in the manuscript, figure legends, and the Description field when uploading your files) using the following format verbatim: S1 Data, S2 Data, etc. Multiple panels of a single or even several figures can be included as multiple sheets in one excel file that is saved using exactly the following convention: S1_Data.xlsx (using an underscore).

b. Deposition in a publicly available repository. Please also provide the accession code or a reviewer link so that we may view your data before publication. 

Fig 1A-B; Fig 2A-B; Fig 3A,C-D; Fig 4A,D; Fig 5A-D; Fig 6A-B; Fig S1A-G; Fig S3; Fig S4; Fig S5A-C; Fig S7A-C; Fig S8A-B

--Please also ensure that figure legends in your manuscript include information on where the underlying data can be found, and ensure your supplemental data file/s has a legend.

--Please ensure that your Data Statement in the submission system accurately describes where your data can be found.

We expect to receive your revised manuscript within two weeks. 

*Published Peer Review History*

*Press*

Sincerely,

Luke

Lucas Smith, Ph.D.

Associate Editor,

lsmith@plos.org,

PLOS Biology

Reviewer remarks:

Reviewer #1, Qun He (note, reviewer 1 has signed this review): This manuscript is acceptable for publication.

Reviewer #2: The authors have addressed most of my comments in the revised version.

I have two final points that should be addressed:

Figure 3: The figure legend needs to be adjusted to the new figure. Please check.

regarding PKA and RCO-1/RCM-1: In my original comment I asked to briefly mention how PKA regulates RCO-1. That was an error on my part. I meant to ask the authors to mention how PKA regulates RCO-1/RCM-1. However, the authors are mistaken that "PKA activates RCO-1/RCM-1...". The opposite is the case: Liu et al. 2015 found that "PKA regulates frq transcription by inhibiting RCM-1 activity through RCM-1 phosphorylation", which is good because this is compatible with their other results. Please check and revise.

Reviewer #3: The authors have addressed my concerns satisfactorily.

---

## [Editor Report · Decision Letter 3]

15 Dec 2022

Dear Dr Kelliher,

Thank you for the submission of your revised Research Article "Nutritional compensation of the circadian clock is a conserved process influenced by gene expression regulation and mRNA stability" for publication in PLOS Biology. Your revised manuscript has now been evaluated by the PLOS Biology editorial team and we are satisfied by the changes made. Therefore, on behalf of my colleagues and the Academic Editor, Gad Asher, I am pleased to say that we can in principle accept your manuscript for publication, provided you address any remaining formatting and reporting issues. These will be detailed in an email you should receive within 2-3 business days from our colleagues in the journal operations team; no action is required from you until then. Please note that we will not be able to formally accept your manuscript and schedule it for publication until you have completed any requested changes.

PRESS

Sincerely, 

Luke

Lucas Smith, Ph.D.

Associate Editor

PLOS Biology

lsmith@plos.org